# Dynamic Optimal Power Flow of Active Distribution Network Based on LSOCR and Its Application Scenarios

**Weiqi Meng** [1] , **Dongran Song** [1] , **Xiaofei Deng** [2,*], **Mi Dong** [1] , **Jian Yang** [1] , **Rizk M. Rizk-Allah** [3,4] **and Václav Snášel** [4]

1 School of Automation, Central South University, Changsha 410083, China
2 School of Information Technology and Management, Hunan University of Finance and Economics, Changsha 410205, China
3 Department of Basic Engineering Science, Faculty of Engineering, Menoufia University, Shebin El-Kom 32511, Egypt
4 Faculty of Electrical Engineering and Computer Science, VŠB-Technical University of Ostrava, 70800 Ostrava, Czech Republic
* Correspondence: dengxiaofei@hufe.edu.cn; Tel.: +86-139-7434-1334

**Abstract:** Optimal power flow (OPF) is a crucial aspect of distribution network planning and operation. Conventional heuristic algorithms fail to meet the system requirements for speed and accuracy, while linearized OPF approaches are inadequate for distribution networks with high R/X ratios. To address these issues and cater to multi-period scenarios, this study proposes a dynamic linearized second-order cone programming-based (SOCP) OPF model. The model is built by first establishing a dynamic OPF model based on linearized second-order conic relaxation (LSOCR-DOPF). The components of the active distribution network, such as renewable energy power generation units, energy storage units, on-load-tap-changers, static var compensators, and capacitor banks, are then separately modeled. The model is implemented in MATLAB and solved by YALMIP and GUROBI. Finally, three representative scenarios are used to evaluate the model accuracy and effectiveness. The results show that the proposed LSOCR-DOPF model can ensure calculation time within 3 min, voltage stability, and error control within $10^{-6}$ for all three applications. This method has strong practical value in the fields of active distribution network day-ahead dispatch, accurate modeling of ZIP load, and real-time operation.

**Keywords:** optimal power flow (OPF); active distribution network; linearized second-order conic relaxation (LSOCR); network reconfiguration; ZIP load

## 1. Introduction

Recently, researchers have shown an increased interest in the active distribution network. The integration of various distributed generations, energy storage units, and active management devices has presented new challenges to the planning and operation of distribution networks [1], especially in the field of active management (AM) of distribution networks [2]. It is particularly urgent to develop optimization algorithms and high-performance computing tools applicable to various fields of active distribution networks. Ref. [3] analyzed three kinds of optimization problems of the smart grid: optimal power flow (OPF), unit commitment, and operation planning. Their essence is distribution network optimization, while having different optimization scales. The OPF is of great significance in the development process of the distribution network, and is the most common and fundamental optimization problem in power systems [4]. Research on distribution network OPF has mainly focused on the alternating current power flow (AC-OPF). Exploring a solution method to enhance the solution speed of distribution network AC-OPF while ensuring its optimal operation and fulfilling the requirements of active distribution network planning and operation has been a major concern in the field of power system

research. As a non-convex optimization problem, the OPF is difficult to solve. It is easy to fall into local optimum in the process of solving and has been proven to be an NP-hard problem [5,6]. The power flow constraints are characterized by nonlinearity, and hence the essence of OPF lies in nonlinear programming. The methods for solving OPF problems can be broadly categorized into the following three categories:

(1) Pursuit of local optimal solutions, including early classical methods [7] (such as Simplified Gradient Method, Newton Method, Sequential Quadratic Programming, Interior Point Method) and recent rapidly-developing heuristic algorithms [3,8];

(2) Approximation of power flow equality constraints. For example, the AC-OPF constraints can be approximately linearized as direct current power flow constraints, and the resultant direct current optimal power flow (DC-OPF) problem can be accordingly solved [9];

(3) Relaxation of the power flow equality constraint using convex relaxation techniques [10].

Methods of seeking local optimal solutions, due to their advantages such as simplicity and ease in simulating complex constraints, have been widely applied in solving nonlinear programming models [11]. However, due to the non-convex nature of the OPF problem, these methods cannot guarantee the quality of the solution, and it is impossible to measure the gap between the locally optimal solution and the globally optimal solution. Methods that approximate the power flow equation constraints, such as the DC-OPF, present several obvious disadvantages. Firstly, they can be challenging to apply to research areas related to voltage and reactive power, as well as distribution networks with high R/X ratios. Secondly, the optimal solution of the DC-OPF problem may not be a feasible solution of the original OPF problem, leading to the need for constant adjustment of the tightness of DC-OPF constraints and the need to solve again during the actual optimization process [12]. With the numerous issues associated with seeking local optima and approximating the current flow equality constraints, it requires new methods to deal with the current flow equality constraints.

The convex relaxation technique has gained significant attention in recent years due to its advantages and potential in solving OPF problems in the field of power system optimization. The use of convex relaxation techniques, particularly second-order cone programming (SOCP) relaxation, has become increasingly prevalent. The SOCP mainly converts the original model into a convex programming form, thus obtaining the globally optimal solution and a good computational speed. Refs. [13,14] systematically established a branch flow model (BFM) based on Distflow [15] to solve the OPF model framework, and presented two relaxation steps: (1) Elimination of voltage and current phase angles; (2) Second Order Conic Relaxation (SOCR). The authors also demonstrated the relaxation accuracy of SOCR. As a typical representative of SOCP, SOCR can be summarized in the mathematical field as the classic "dimensionality relaxation-return mapping" process: A new variable is introduced to elevate the dimensionality of the original problem; then, in the elevated problem, non-convex constraints are relaxed and a solution is obtained; finally, the solution of the original problem is recovered through a return mapping. The main challenge in using SOCR to solve the optimal power flow lies in accurately satisfying the conditions of the relaxed model. Refs. [16,17] provided a comprehensive summary of the sufficient conditions for the accuracy of SOCR under radiated network conditions, which are divided into three categories: power injection constraints, voltage amplitude constraints, and node voltage phase angle deviations, and corresponding explanations are given for the optimal power flow objective function. After the examination of Ref. [17], further research on the exact relaxation sufficient condition has been extensively conducted. Ref. [18] expanded upon the sufficient conditions for the accuracy of SOCR in the presence of the high penetration of distributed generation. Refs. [19,20] highlighted the shortcomings of the second sufficient condition in Ref. [17], as it neglected the impact of the grounding branch power flow on line capacity, and proposed an improved sufficient condition.

Additionally, although the OPF model of the active distribution network based on MISOCP has achieved a relatively high solution efficiency [21], strictly speaking, it is still

a nonlinear model, and the efficiency of solving nonlinear second-order cone constraints will decrease with the increase in the number of distribution network nodes. In order to achieve efficient solving of the OPF problem in future large-scale distribution networks, it is necessary to simplify the second-order cone programming constraints while ensuring the accuracy and efficiency of distribution network solving. Ref. [22] has proven that the second-order cone in dimension N, can be outer-approximated to an arbitrary accuracy by a polyhedral cone in an extended space. Polyhedral approximations are very powerful for solving MISOCP as the benefits of warm-starting for LPs can be utilized throughout the branch-and-bound algorithm [23]. This is particularly important for problems that can be formulated as MISOCP, such as the OPF problem in this paper. Meanwhile, Refs. [24,25] extended the constant power load model to a ZIP load model under a rectangular coordinate system and applied it to the solution of the OPF. Meanwhile, most of the existing distribution network models only include power generation units and energy storage devices, and rarely consider reactive power compensation devices and other active management devices at the same time [26]. In addition, in the active distribution network planning and operation optimization model, discrete variables will inevitably appear with the active management devices considered, which turns the original problem into a mixed integer linear programming (MILP) [27]. With continuous development and maturity, commercial optimization software (GUROBI, CPLEX, MOSEK, etc.) has been widely used in distribution network reconstruction [28], reactive power optimization [29], distribution network planning [11], etc.

It is noteworthy that the above studies were mostly limited to the traditional single-period static OPF category, while the actual optimization requires the overall coordination of multi-periods, which is actually a dynamic optimal power flow (DOPF). Additionally, compared to the rectangular coordinate system, the current form in polar coordinates is more common. Therefore, in order to meet the operational requirements of multi-period scenarios, various complex constraints, and fast large-scale solutions for active distribution networks, this paper proposes a linearized second-order cone relaxation dynamic optimal power flow (LSOCR-DOPF) model of the distribution network and explores the linear modeling method of key constraints of active distribution network participating elements (such as on-load-tap-changer (OLTC), static var compensator (SVC), capacitor banks (CB), the energy storage system (ESS), etc.). The nonlinear OPF problem of the distribution network is transformed into a computationally efficient solution utilizing MILP, with a comprehensive explanation of the network's constraints on radiation and connectivity. On this basis, this paper further presents an innovative approximation of the ZIP load model under polar coordinates and validates the effectiveness of the OPF framework through three scenarios: power coordination optimization, network reconfiguration, and ZIP load application. The main contributions of this study can be summarized as follows:

- Present the LSOCR-DOPF model, which is based on branch power flow analysis, for the active distribution network.
- Explore the linear modeling method for the constraints of various active management units, including OLTC, SVC, CB, ESS, etc.
- Validate the LSOCR-DOPF model through simulation experiments in three typical scenarios: power coordination optimization, network reconfiguration, and ZIP load application.

The article is organized as follows: The Section 2 presents the LSOCR-DOPF model for active distribution networks and the design of various active management units based on branch power flow analysis. The Section 3 discusses the results of three simulation experiments and provides a comprehensive analysis and discussion of the findings. Finally, the Section 4 presents the conclusion and future perspectives.

## 2. Methodology

### 2.1. LSOCR-DOPF Model of the Distribution Network

#### 2.1.1. Basic Structure of Distribution Networks

In most distribution networks, the steady-state power operation mode is radial and its structure is depicted in Figure 1. For a radial topology network, the node directed graph can be used for equivalent analysis. Furthermore, $S_{ij}$ and $S_i$ represent complex power, $S_{ij} = P_{ij} + Q_{ij}i$ and $S_i = p_i + q_i i$. Branch complex impedance $Z_{ij} = r_{ij} + x_{ij}i$. Set $B$ represents the set of all nodes in the network. In the traditional OPF, the voltage remains constant. If OLTC is installed, the voltage will change with the OLTC transformation ratio; E represents the collection of all branches in the network. There are $N^{\text{sub}}$ substations, $N^{\text{bus}}$ nodes, and $N^{\text{Line}}$ line branches in the network.

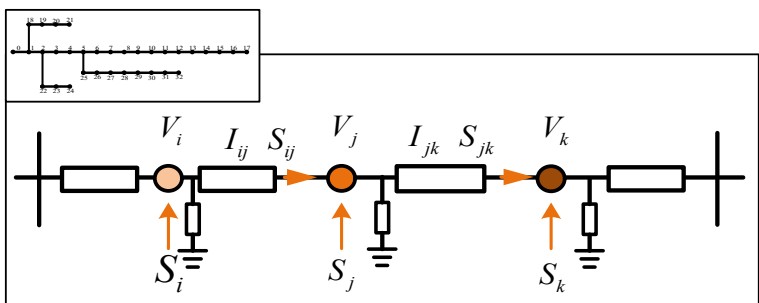

**Figure 1.** Structure of radial distribution network.

#### 2.1.2. Basic OPF Model Based on BFM

Generally, the basic model of optimal power flow based on branch power flow (BFM-OPF, Figure 1) is expressed as follows [14]:

$$\min \quad f(p, q, P, Q, V, I) \tag{1}$$

$$\text{s.t.} \begin{cases} p_j = \sum_{k \in \delta(j)} P_{jk} - \sum_{i \in \pi(j)} \left( P_{ij} - I_{ij}^2 r_{ij} \right) + g_j V_j^2, \forall j \in B, \forall ij \in E \\ q_j = \sum_{k \in \delta(j)} Q_{jk} - \sum_{i \in \pi(j)} \left( Q_{ij} - I_{ij}^2 x_{ij} \right) + b_j V_j^2, \forall j \in B, \forall ij \in E \end{cases} \tag{2}$$

$$V_j^2 = V_i^2 - 2\left( P_{ij} r_{ij} + Q_{ij} x_{ij} \right) + I_{ij}^2 \left( r_{ij}^2 + x_{ij}^2 \right), \forall i, j \in B, \forall ij \in E \tag{3}$$

$$I_{ij}^2 = \frac{P_{ij}^2 + Q_{ij}^2}{V_i^2}, \forall j \in B, \forall ij \in E \tag{4}$$

$$\underline{I}_{ij} \leq I_{ij} \leq \overline{I}_{ij}, \forall ij \in E \tag{5}$$

$$\underline{V}_j \leq V_j \leq \overline{V}_j, \forall j \in B \tag{6}$$

$$\begin{cases} p_j \in R_j^p \\ q_j \in R_j^q \end{cases}, \forall j \in B \tag{7}$$

where, $p_j$ and $q_j$ represent the active and reactive power injections, respectively, at each node; the branch $ij$ represents the positive direction of flow direction from node $i$ to node $j$; $\delta(j)$ is the collection of branch end nodes with $j$ as the head node, and $\pi(j)$ is the collection of branch end nodes with $j$ as the end node; $P_{jk}$ and $Q_{jk}$ denote the active and reactive power at the head node of the branch $ij$, respectively; $P_{ij}$ and $Q_{ij}$ denote the active and reactive current flow in each branch; $r_{ij}$ and $x_{ij}$ correspond to the individual resistance and reactance of each branch; $g_j$ and $b_j$ are the separate ground conductance and ground susceptance of

node $j$; $\underline{V}_j$ and $\overline{V}_j$ are the upper and lower limits of node voltage, respectively; $\underline{I}_{ij}$ and $\overline{I}_{ij}$ are the upper and lower limits of branch $ij$ current, respectively.

From Equations (1)–(7), it can be inferred that: (1) The optimization variables for the OPF consist of node injection power $(p, q)$, branch power flow $(P, Q)$, node voltage $(V)$, and branch current $(I)$, with the substation node voltage not considered as an optimization variable. (2) Equation (1) represents the objective function, which can be the minimization of network losses and substation node power purchases. (3) The well-known branch flow equation [30] is expressed by Equations (2) and (3), while Equation (4) represents the power calculation equation. Equations (5) and (6) denote the safety constraint equations for branch currents and nodal voltages, respectively. Equation (7) introduces node-dependent constraints that are subject to change based on the employed model.

2.1.3. Multi-Period LSOCR-OPF Model

In order to convert power flow constraints into quadratic cone constraints, additional variables of cone optimization need to be set:

$$\begin{cases} \overline{I}_{ij} = I_{ij}^2, \forall ij \in E \\ \overline{V}_j = V_j^2, \forall j \in B \end{cases} \tag{8}$$

Substitute Equation (8) into Equations (2)–(6) to change the power flow constraint to the second-order cone constraint [31] as follows:

$$\min \quad f(p, q, P, Q, V, I) \tag{9}$$

$$\text{s.t.} \begin{cases} p_j = \sum_{k \in \delta(j)} P_{jk} - \sum_{i \in \pi(j)} \left( P_{ij} - \overline{I}_{ij} r_{ij} \right) + g_j \overline{V}_j, \forall j \in B, \forall ij \in E \\ q_j = \sum_{k \in \delta(j)} Q_{jk} - \sum_{i \in \pi(j)} \left( Q_{ij} - \overline{I}_{ij} x_{ij} \right) + b_j \overline{V}_j, \forall j \in B, \forall ij \in E \end{cases} \tag{10}$$

$$\overline{V}_j = \overline{V}_i - 2 \left( P_{ij} r_{ij} + Q_{ij} x_{ij} \right) + \overline{I}_{ij} \left( r_{ij}^2 + x_{ij}^2 \right), \forall j \in B, \forall ij \in E \tag{11}$$

$$\left\| \begin{matrix} 2P_{ij} \\ 2Q_{ij} \\ \overline{I}_{ij} - \overline{V}_j \end{matrix} \right\|_2 \leq \overline{I}_{ij} + \overline{V}_j, \forall j \in B, \forall ij \in E \tag{12}$$

$$\underline{I}_{ij}^2 \leq \overline{I}_{ij} \leq \overline{I}_{ij}^2, \forall ij \in E \tag{13}$$

$$\underline{V}_j^2 \leq \overline{V}_j \leq \overline{V}_j^2, \forall j \in B \tag{14}$$

where, $\| \ \|_2$ represents the L2 norm.

With this, the SOCR-OPF (a typical Mixed-Integer Second Order Cone Programming, MISOCP) model is fully modeled. Among them, the second-order cone programming represented by Equation (12) can be expanded and simplified as:

$$\sqrt{P_{ij}^2 + Q_{ij}^2} \leq \overline{V}_j \overline{I}_{ij}, \forall j \in B, \forall ij \in E \tag{15}$$

Equation (15) can be uniformly described as:

$$\sqrt{x_1^2 + x_2^2} \leq x_3; x_1 = P_{ij}, x_2 = Q_{ij}, x_3 = \sqrt{\overline{V}_j \overline{I}_{ij}}, \forall j \in B, \forall ij \in E \tag{16}$$

Ben-Tal and Nemirovski [22] showed that Equation (16) could be approximated by a system of linear homogeneous equalities and inequalities in terms of $x_1, x_2, x_3$, and $2(k + 1)$

variables $a_k, b_k$ for $k = 0, \ldots, K$. $K$ is a parameter of the polyhedral-relaxed approximation. The approximate expression of the polyhedron of the three-dimensional SOCR constraint (Equation (16)) is:

$$\begin{cases} a_0 \geqslant |x_1|; b_0 \geqslant |x_2|; a_k = a_{k-1}\cos\frac{\pi}{2^{k+1}} + b_{k-1}\sin\frac{\pi}{2^{k+1}} \\ \qquad b_k \geqslant \left| -a_{k-1}\sin\frac{\pi}{2^{k+1}} + b_{k-1}\cos\frac{\pi}{2^{k+1}} \right| \qquad , k = 1, 2, \cdots, K \\ \qquad a_K \leqslant x_3; b_K \leqslant a_K \tan\frac{\pi}{2^{k+1}} \end{cases} \qquad (17)$$

where, $K$ represents the number of facets in a polyhedron. $a_0$ and $b_k$ are auxiliary variables.

The polyhedral approximation given by (17) can be reduced by using the linear equality constraints $a_k = a_{k-1}\cos\left(\pi/2^{k+1}\right) + b_{k-1}\sin\left(\pi/2^{k+1}\right)$ to solve for $a_k(k = 0, \ldots, K)$ in terms of $a_0$ and $a_k(k = 0, \ldots, K)$ and then substitute $a_k$ out of Equation (17). The resulting system will only have linear inequality constraints in terms of $x_1, x_2, x_3, a_0$, and the $(k + 1)$ variables $b_k$ for $k = 0, \ldots, K$. The error ($\varepsilon(K)$) [22] of the polyhedral approximations of the three-dimensional second-order cone constraint (Equation (16)) is:

$$\varepsilon(K) = \frac{1}{\cos\left(\frac{\pi}{2^{K+1}}\right)} - 1 \qquad (18)$$

According to Equation (18), when $K = 11$, the error is about $3 \times 10^{-7}$. Therefore, the MISOCP model for active distribution network reconfiguration is approximately equivalent to the MILP model. At this point, the improved version of SOCR-OPF, namely the LSOCR-OPF model, is fully modeled.

So far, the power flow constraint has been transformed from a nonlinear programming model to a MISOCP model composed of Equations (9)–(14). Then, through the polyhedron approximation method, which is used to linearize Equation (16), the MISOCP model can be approximately converted into a MILP model for solution.

The previously described model represents the conventional single-period OPF model. However, since most practical applications involve multi-period optimization, this study converts the single-period (static) power flow model into a multi-period (dynamic) OPF model. For clarity of expression, the following dynamic OPF model, based on LSOCR, can be expressed in vector form:

$$\begin{cases} min \quad \sum_{t \in T} f(x_t) \\ \text{s.t.} \quad x_t \in X_t, \forall t \\ \qquad A_{ij,t}x_t \leq b_{ij,t}^{\mathrm{T}}x_t, \forall t, \forall ij \in E \\ \qquad \sum_{t \in T} B_t x_t \leq c \\ \qquad \sum_{t \in T} C_t x_t = d \end{cases} \qquad (19)$$

where, $t$ is the period identification, $T$ is the total number of time periods. $x_t \in X_t, \forall t$ represents the constraint relationship in the traditional single-period OPF model, such as upper and lower limit constraints, power flow equation constraints, etc. $\sum_{t \in T} B_t x_t \leq c$ is the second-order cone constraint relationship under each branch at each time.

Equation (19) adds the linear coupling relationship between multi-period periods to the objective function and constraint conditions. Some elements will be described in detail in the next section, such as OLTC, CB, ESS, etc. Hence, the further improved version of LSOCR-OPF, namely the LSOCR-DOPF model, is fully modeled.

In order to facilitate understanding, additional explanation is required for the entire model transformation process (from AC-OPF to LSOCR-DOPF): Equations (8) and (16) respectively embody the phase angle relaxation and second-order cone relaxation of LSOCR-

DOPF, and Figure 2 depicts the schematic diagram of the two-step relaxation process. The non-convex feasible region $C_{AC-OPF}$ of the original AC-OPF problem will be relaxed into a convex second-order cone feasible region $C_{OPF-cr}$ after phase angle relaxation and second-order cone relaxation. Then, the convex feasible region of the second-order cone is further linearized into the convex feasible region $C_{OPF-linear}$ of the integer programming by the polyhedral approximations. At this time, the optimal power flow problem in the original formulation has already been transformed into a convex optimization problem. Numerous studies, as demonstrated in Refs. [13,14,16,17], have substantiated the strict accuracy of the second-order cone relaxation (SOCR) approach for most distribution network structures, when the objective function is both a convex and strictly increasing function.

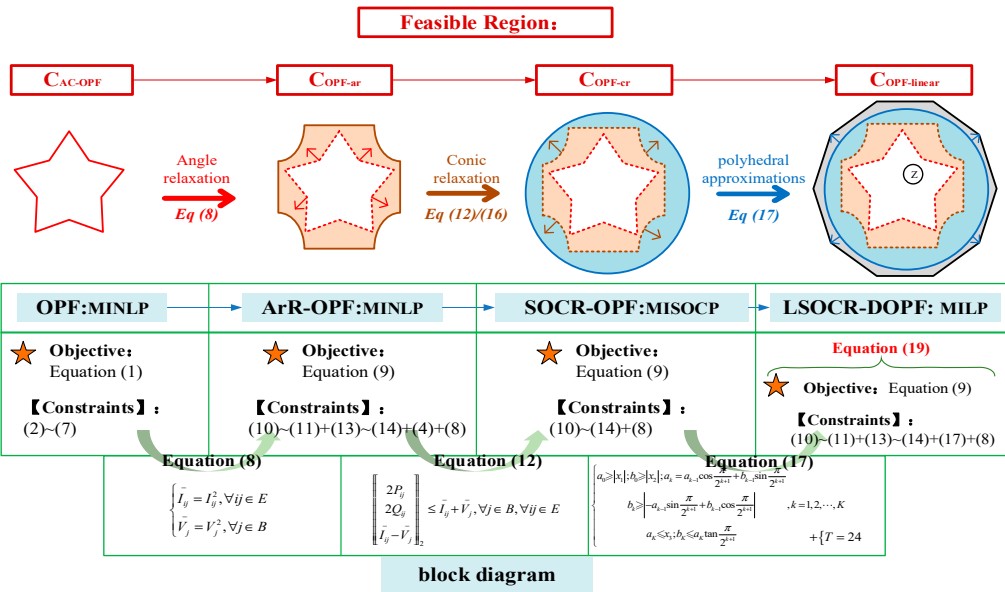

**Figure 2.** The proposed modeling process of LSOCR for solving AC-OPF.

## 2.2. Active Distribution Network Modeling

In this section, the active management components of the active distribution network are considered, including: ① reactive compensation device (SVC and CB); ② active power regulation device (energy storage system and electric vehicle mobile energy storage system); ③ on-load tap changer (OLTC); ④ distributed generation power regulation. Given that the objective function in the OPF is either linear or quadratic, the quadratic form can be effectively addressed by means of piecewise linearization. This paper linearizes the constraints related to the active management equipment. Given that the network reconfiguration is of great significance for active distribution network planning and optimal operation [32], this paper discusses the constraints related to grid reconfiguration, such as radial constraints. In addition, in order to ensure the applicability of the model, this paper further considers the ZIP load.

### 2.2.1. Active Distribution Network Units Modeling

1. Active power regulation device

(1) Modeling of discrete reactive power compensation (CB).

$$\begin{cases} Q_{j,t}^{CB} = y_{j,t}^{CB} Q_j^{CB,step} \\ y_{j,t}^{CB} \le Y_j^{CB,m} \end{cases}, \forall t, \forall j \in B^{CB} \tag{20}$$

where, $B^{CB}$ is the set of CB nodes; $y_{j,t}^{CB}$ is the number of groups put into operation and the discrete variable value; $Y_j^{CB,max}$ is the upper limit of the

number of CB groups connected by node $j$; $Q_j^{\text{CB, step}}$ is the compensation power of each group of CB. Considering factors such as equipment life or economy, discrete reactive compensation is mostly limited by the number of adjustments, so it generally includes the total number of operations in multiple periods; $N_j^{\text{CB,max}}$ is the upper limit of operation times:

$$\sum_{t \in T} \left| y_{j,t}^{\text{CB}} - y_{j,t-1}^{\text{CB}} \right| \leq N_j^{\text{CB},m}, \forall t, \forall j \in B^{\text{CB}} \tag{21}$$

In addition, for the absolute value constraint in the above equation, add an auxiliary variable $\delta_{j,t}^{\text{CB}} = \left| y_{j,t}^{\text{CB}} - y_{j,t-1}^{\text{CB}} \right|$ that represents the change in CB compensation capacity between adjacent periods, corresponding to:

$$\begin{cases} \sum_{t \in T} \delta_{j,t}^{\text{CB}} \leq N_j^{\text{CB},m} \\ -\delta_{j,t}^{\text{CB}} Y_j^{\text{CB},m} \leq y_{j,t}^{\text{CB}} \leq \delta_{j,t}^{\text{CB}} Y_j^{\text{CB},m} \end{cases}, \forall t, \forall j \in B^{\text{CB}} \tag{22}$$

(2)    Modeling of continuous reactive power regulation device (SVC).

$$Q_j^{\text{SVC,min}} \leq Q_{j,t}^{\text{SVC}} \leq Q_j^{\text{SVC},m}, \forall t, \forall j \in B^{\text{SVC}} \tag{23}$$

where, $B^{\text{SVC}}$ is the node set containing SVC; $Q_j^{\text{SVC, min}}$ and $Q_j^{\text{SVC,max}}$ are the lower limit and upper limit of SVC compensation power, respectively. Considering that, in the process of active distribution network operation, with the increasing penetration of distributed generation (DG) such as photovoltaic power generation, the system power flow may be reversed and overvoltage problems may occur, the lower limit $Q$ of SVC compensation in this paper is: $Q_j^{\text{SVC,min}} < 0$.

2.  OLTC model
    The OLTC is used to adjust the voltage value at the low-voltage side of the bus node. Therefore, the substation bus node $V_0$ is further converted to the adjustable variable:

$$\begin{cases} V_j^2 \leq \left( V_{j,t}^{\text{Base}} \right)^2 r_{j,t} \leq \overline{V}_j^2 \\ r_j^{\min} \leq r_{j,t} \leq r_j^{\max} \end{cases}, \forall t, \forall j \in B^{\text{OLTC}} \tag{24}$$

where, $B^{\text{OLTC}}$ refers to the node set of substation containing OLTC; $V_{j,t}^{\text{Base}}$ is the voltage value at the high voltage side of the transformer, which is a constant value; $r_j^{\max}$ and $r_j^{\min}$ are the square of the upper and lower limit of the OLTC adjustable transformation ratio; $r_{j,t}$ is the square of the OLTC transformation ratio, defined as the ratio of the secondary side to the primary side, which is actually a discrete value variable, and can be further treated as the following relationship including 0–1 variables:

$$r_{j,t} = r_j^{\min} + \sum_s r_{j,s} \sigma_{j,s,t}^{\text{OLTC}}, \forall t, \forall j \in B^{\text{OLTC}} \tag{25}$$

where, $r_{j,s}$ represents the difference between OLTC gear $s$ and the square of gear $s - 1$, which is the adjacent adjustment increment. $\sigma_{j,s,t}^{\text{OLTC}}$ is a 0–1 identification variable. If

it is considered to be constrained by the limit of adjustment times in practice, it can be further constrained as:

$$
\text{OLTC}\begin{cases}
\sigma_{j,1,t}^{\text{OLTC}} \geq \sigma_{j,2,t}^{\text{OLTC}} \geq_{j,\text{SR}_j,t}^{\text{OLTC}} \\
\delta_{j,t}^{\text{OLTC,IN}} + \delta_{j,t}^{\text{OLTC,DE}} \leq 1 \\
\sum_s \sigma_{j,s,t}^{\text{OLTC}} - \sum_s \sigma_{j,s,t-1}^{\text{OLTC}} \geq \delta_{j,t}^{\text{OLTC,IN}} - \delta_{j,t}^{\text{OLTC,DE}} \text{SR}_j \\
\sum_s \sigma_{j,s,t}^{\text{OLTC}} - \sum_s \sigma_{j,s,t-1}^{\text{oLTC}} \leq \delta_{j,t}^{\text{OLTC,IN}} \text{SR}_j - \delta_{j,t}^{\text{OLTC,DE}} \\
\sum_{t \in T}\left(\delta_{j,t}^{\text{OLTC,N}} + \delta_{j,t}^{\text{OLTC,DE}}\right) \leq N_j^{\text{OLTC,max}}
\end{cases} \quad ,\forall t, \forall j \in B^{\text{OLTC}} \quad (26)
$$

where, $\delta_{j,t}^{\text{OLTC,N}}$ and $\delta_{j,t}^{\text{OLTC,DE}}$ represent the OLTC gear adjustment change sign, which is 0–1 variable; if $\delta_{j,t}^{\text{OLTC,IN}} = 1$, then the gear value of OLTC at time $t-1$ is greater than the gear value at time $t$, $\delta_{j,t}^{\text{OLTC,DE}}$ is similar; $\text{SR}_j^{\text{OLTC}}$ is the maximum range of gear change; $N_j^{\text{OLTC, max}}$ is the maximum allowable adjustment times of the OLTC gear at time $T$.

3.  ESS model
    During this part, the modeling of the energy storage system takes into account multiple-period constraints, including restrictions on its charging and discharging status, charging and discharging power, as well as capacity limitations.

    (1)  Power limit.

$$
\begin{cases}
u_{j,t}^{\text{discharge}} P_j^{\text{discharge,min}} \leq P_{j,t}^{\text{discharge}} \leq u_{j,t}^{\text{discharge}} P_j^{\text{discharge,max}} \\
u_{j,t}^{\text{charge}} P_j^{\text{charge,min}} \leq P_{j,t}^{\text{charge}} \leq u_{j,t}^{\text{charge}} P_j^{\text{charge,max}}
\end{cases} \quad ,\forall t, \forall j \in B^{\text{ESS}} \quad (27)
$$

    (2)  Charge and discharge status limit.

$$
u_{j,t}^{\text{discharge}} + u_{j,t}^{\text{charge}} \leq 1, \forall j \in B^{\text{ESS}}, \forall t \quad (28)
$$

    (3)  Capacity constraints.

$$
\begin{cases}
E_{j,t+1}^{\text{ESS}} = E_{j,t}^{\text{ESS}} + \alpha_j^{\text{charge}} P_{j,t}^{\text{charge}} - \alpha_j^{\text{discharge}} P_{j,t}^{\text{discharge}} \\
E_j^{\text{ESS,min}} \leq E_{j,t}^{\text{ESS}} \leq E_j^{\text{ESS,max}}
\end{cases} \quad ,\forall j \in B^{\text{ESS}}, \forall t \quad (29)
$$

    where, $B^{\text{ESS}}$ is the node set containing ESS; Equation (23) indicates that the ESS cannot be simultaneously charged and discharged at the same time, $u_{j,t}^{\text{charge}}$ and $u_{j,t}^{\text{discharge}}$ is $P_j^{\text{charge, min}}$ and $P_j^{\text{discharge,min}}$ are the upper and lower limits of ESS charging and discharging power, respectively; $E_{j,t}^{\text{ESS}}$ is the power of the t period of ESS, $E_j^{\text{ESS,max}}$ and $E_j^{\text{ESS, min}}$ are the upper and lower limit values considering factors such as ESS life; $\alpha_j^{\text{charge}}$ and $\alpha_j^{\text{discharge}}$ are the charge and discharge efficiency coefficient, respectively, generally $0 < \alpha_j^{\text{charge}} < 1$, $\alpha_j^{\text{discharge}} > 1$. As a novel active management technique, electric vehicles can be considered as mobile active power energy storage systems [33]. Their basic model is largely similar to that of Energy Storage Systems (ESS). The equivalent injection power at each bus node in the distribution network can be represented as the clustering outcomes of individual electric vehicles.

4.  Distributed generation model
    Respectively modeling DG with or without reactive power:

    (1)  DG modeling without considering reactive power.

Currently, the modeling form of active management for DG primarily takes into account the possibility of DG allowing power shedding under certain conditions, and assumes that DG is only related to active power output [34], that is:

$$0 \leq P_{j,t}^{\mathrm{DG}} \leq P_{j,t}^{\mathrm{DG,PRE}}, \forall t, \forall j \in B^{\mathrm{DG}} \tag{30}$$

where, $B^{\mathrm{DG}}$ is the node set containing DG; $P_{j,t}^{\mathrm{DG,\ PRE}}$ is the predicted active power output of node $j$ at time $t$.

(2)　DG modeling considering reactive power [35].

With the maturity of active power regulation as a primary method in the study of DG and the advent of new technology, some DGs can have a certain impact on reactive power in the power grid, including outputting and absorbing reactive power. As a result, active management research focusing on the reactive power of DG has emerged, mainly divided into constant power factor control and variable power factor control.

①　Constant power factor control

$$\begin{cases} 0 \leq P_{j,t}^{\mathrm{DG}} \leq P_{j,t}^{\mathrm{DG,PRE}} \\ Q_{j,t}^{\mathrm{DG}} = F_j^{\mathrm{DG}} P_{j,t}^{\mathrm{DG}} \end{cases}, \forall t, \forall j \in B^{\mathrm{DG}} \tag{31}$$

②　Variable power factor control

$$\begin{cases} 0 \leq P_{j,t}^{\mathrm{DG}} \leq P_{j,t}^{\mathrm{DG,PRE}} \\ Q_{j,t}^{\mathrm{DG}} = F_{j,t}^{\mathrm{DG}} P_{j,t}^{\mathrm{DG}} \\ F_j^{\mathrm{DG,min}} \leq F_{j,t}^{\mathrm{DG}} \leq F_j^{\mathrm{DG},m}, \forall t, \forall j \in B^{\mathrm{DG}} \\ Q_j^{\mathrm{DG,min}} \leq Q_{j,t}^{\mathrm{DG}} \leq Q_j^{\mathrm{DG,max}} \end{cases} \tag{32}$$

where, $F_j^{\mathrm{DG,\ max}}$ and $F_j^{\mathrm{DG\ min}}$ are the upper and lower limits of DG transformation ratio adjustment; $F_{j,t}^{\mathrm{DG}}$ represents the ratio of reactive power to active power, and its optimization range can be converted from the power factor control range. At present, most of the literature will limit the reactive power $Q_j^{\mathrm{DG,min}}$ and $Q_j^{\mathrm{DG,max}}$ is set as a constant [27].

### 2.2.2. Radial Constraints

Network reconfiguration is of great significance for the active distribution network with high renewable energy penetration, and radiation and islanding constraints are more important constraints in the process of the optimal operation of the distribution network. Since the BFM model assumes the positive direction of the power flow, that is, $P_{ij} > 0$ indicates that the current flows from $i$ to $j$, if $P_{ij} < 0$, the current direction is opposite. In addition, suppose $\sigma_{ij}$ (0–1 variable) represents the state of branch $ij$, if $\sigma_{ij,t} = 0$, then the branch $ij$ switch is open, and vice versa. It can be obtained in the same way in the distribution network planning, if $\sigma_{ij,t} = 1$, then the candidate branch $ij$ is connected, and vice versa. Based on this method, the LSOCR-DOPF model in this paper needs to add the following constraints:

$$\begin{cases} P_{ij}^{\min} \sigma_{ij,t} \leq P_{ij,t} \leq P_{ij}^{\max} \sigma_{ij,t} \\ Q_{ij}^{\min} \sigma_{ij,t} \leq Q_{ij,t} \leq P_{ij}^{\max} \sigma_{ij,t}, \forall ij \in E^{\mathrm{SW}} \\ \overline{I}_{ij}^{\min} \sigma_{ij,t} \leq \overline{I}_{ij,t} \leq \overline{I}_{ij}^{\max} \sigma_{ij,t} \\ \sum_{ij \in E/E^{\mathrm{sw}}}'' 1'' + \sum_{ij \in E^{\mathrm{sw}}} \sigma_{ij,t} = N^{\mathrm{bus}} - N^{\mathrm{sub}} \end{cases} \tag{33}$$

Equation (33) imposes a stringent restriction on the branch power flow. It serves as both necessary and sufficient conditions, which guarantee a certain level of load power balance. In a purely load-based grid, these conditions can ensure stability. However, in an active distribution network integrating DERs, there may arise the possibility of ring networks or isolated islands. The introduction of connectivity constraints can effectively eliminate such isolated island operations in the power grid. This can be achieved by setting the injection power of all non-substation nodes in the network to a small normal value $\varepsilon$ and incorporating it into the power flow constraint, ensuring connectivity between each node and the substation node. The corresponding auxiliary equation is as follows:

$$
\begin{cases}
p_{j,t}^* = \sum\limits_{k \in \delta(j)} P_{jk,t}^* - \sum\limits_{i \in \pi(j)} \left( P_{ij,t}^* - \bar{I}_{ij,t}^* r_{ij} \right) + g_j \overline{V}_{j,t}^* = \varepsilon \\
q_{j,t}^* = \sum\limits_{k \in \delta(j)} Q_{jk,t}^* - \sum\limits_{i \in \pi(j)} \left( Q_{ij,t}^* - \bar{I}_{ij,t}^* x_{ij} \right) + b_j \overline{V}_{j,t}^* = \varepsilon \\
\overline{V}_{j,t}^* = \overline{V}_{j,t}^* - 2 \left( P_{ij,t}^* r_{ij} + Q_{ij,t}^* x_{ij} \right) + \bar{I}_{ij,t}^* \left( r_{ij}^2 + x_{ij}^2 \right), \forall j \in B, \forall ij \in E
\end{cases}
\tag{34}
$$

So far, Equations (33) and (34) ensure the radiation and connectivity in the network. At the same time, in the process of power grid reconstruction, the number of switch adjustments is also limited. Similar to the constraint on the number of OLTC adjustments in Section 2.2.1, the switch adjustment limit is modeled as follows:

$$
\begin{cases}
\delta_{ij,t}^{\text{SW,IN}} + \delta_{ij,t}^{\text{SW,DE}} \leq 1 \\
\sigma_{ij,t} - \sigma_{ij,t-1} \geq \delta_{ij,t}^{\text{SW,IN}} - \delta_{ij,t}^{\text{SW,DE}} \quad , \forall t, \forall ij \in E^{\text{SW}} \\
\sigma_{ij,t} - \sigma_{ij,t-1} \leq \delta_{ij,t}^{\text{SW,IN}} - \delta_{ij,t}^{\text{SW,DE}}
\end{cases}
\tag{35}
$$

where, $\delta_{ij,t}^{\text{sw,I}}$ and $\delta_{ij,t}^{\text{sw,DE}}$ represent the branch switch change identification, which is 0–1 variable, if $\delta_{ij,t}^{\text{SW,IN}} = 1$, then the switch changes from open state to closed state at time $t$, $\delta_{ij,t}^{\text{sw,DE}}$ can be obtained in the same way; $N_{ij}^{\text{sW,max}}$ is the maximum allowable adjustment times of the switch at time $T$.

### 2.2.3. ZIP Load Application

It is evident that the traditional approach of only considering constant power loads in demand is no longer adequate in the era of advanced technology and the growing demand for refined simulation. Thus, it is imperative to develop a ZIP load modeling that accounts for voltage static characteristics. As noted in [25], the load can be modeled using the ZIP model, which is comprised of constant power, constant current, and constant impedance models. For ease of expression, the time marker $t$ has been omitted.

$$
\begin{cases}
P_j^{\text{Load}} = P_{j,0}^{\text{load}} \left( \alpha_j^{\text{A}} \dfrac{V_j^2}{V_{j,0}^2} + \alpha_j^{\text{B}} \dfrac{V_j}{V_{j,0}} + \alpha_j^{\text{C}} \right) \\
Q_j^{\text{Load}} = Q_{j,0}^{\text{load}} \left( \alpha_j^{\text{A}} \dfrac{V_j^2}{V_{j,0}^2} + \alpha_j^{\text{B}} \dfrac{V_j}{V_{j,0}} + \alpha_j^{\text{C}} \right)
\end{cases}
\tag{36}
$$

where, $\alpha_j^{\text{A}}$, $\alpha_j^{\text{B}}$, and $\alpha_j^{\text{C}}$ are the proportion of constant impedance load, constant current load, and constant power load; $P_{j,0}^{\text{load}}$, $Q_{j,0}^{\text{load}}$ are the power demand under the rated voltage. According to the second-order cone relaxation principle, the model is equivalent to:

$$
\begin{cases}
P_j^{\text{Load}} = P_{j,0}^{\text{load}} \left( \alpha_j^{\text{A}} \dfrac{\overline{V}_j}{\overline{V}_{j,0}} + \alpha_j^{\text{B}} \dfrac{\sqrt{\overline{V}_j}}{\sqrt{\overline{V}_{j,0}}} + \alpha_j^{\text{C}} \right) \\
Q_j^{\text{Load}} = Q_{j,0}^{\text{load}} \left( \alpha_j^{\text{A}} \dfrac{\overline{V}_j}{\overline{V}_{j,0}} + \alpha_j^{\text{B}} \dfrac{\sqrt{\overline{V}_j}}{\sqrt{\overline{V}_{j,0}}} + \alpha_j^{\text{c}} \right)
\end{cases}
\tag{37}
$$

where, for constant current load, $\sqrt{\overline{V}_j}$, since the unit value of voltage is near the rated voltage of 1.0, the square term is also near the rated value of 1.0, set $\overline{V}_j = 1 + \Delta \overline{V}_j$, $\left| \Delta \overline{V}_j \right| < 0.1$. By expanding it in Taylor series:

$$\sqrt{\overline{V}_j} = \sqrt{1 + \Delta \overline{V}_j} = 1 + \frac{1}{2}\Delta \overline{V}_j + o\left(\Delta \overline{V}_j^2\right) \tag{38}$$

It should be noted that:

$$\sqrt{\overline{V}_j} \approx 1 + \frac{1}{2}\Delta \overline{V}_j = \frac{1}{2} + \frac{1}{2}\overline{V}_j \tag{39}$$

$$\Psi\left(\overline{V}_j\right) = \left| \sqrt{\overline{V}_j} - \left(\frac{1}{2} + \frac{1}{2}\overline{V}_j\right) \right| \tag{40}$$

The calculated equivalent error order is $10^{-3}$, which fully meets the optimization requirements. Based on the above analysis, $\overline{V}_{j,0} = \sqrt{\overline{V}_{j,0}} = 1$ is substituted into Equation (37) to obtain:

$$\begin{cases} P_j^{\text{Load}} = \left(\alpha_j^A + \frac{\alpha_j^B}{2}\right)\overline{V}_j P_{j,0}^{\text{load}} + \left(\frac{\alpha_j^B}{2} + \alpha_j^C\right) P_{j,0}^{\text{load}} \\ Q_j^{\text{Load}} = \left(\alpha_j^A + \frac{\alpha_j^B}{2}\right)\overline{V}_j Q_{j,0}^{\text{load}} + \left(\frac{\alpha_j^B}{2} + \alpha_j^C\right) Q_{j,0}^{\text{load}} \end{cases} \tag{41}$$

### 2.3. Proposed LSOCR-DOPF Optimization Flowchart

The comprehensive optimization flowchart of LSOCR-DOPF is depicted in Figure 3. The comprehensive LSOCR-DOPF model described in this paper is comprised of the preliminary LSOCR-DOPF model (Equation (19)) introduced in Section 2.1.3 and all of the linearized models outlined in Section 2.2. The LSOCR-DOPF is a classic Mixed Integer Linear Programming (MILP) problem, which can be modeled using the YALMIP toolkit in MATLAB and can be efficiently solved by invoking advanced solvers.

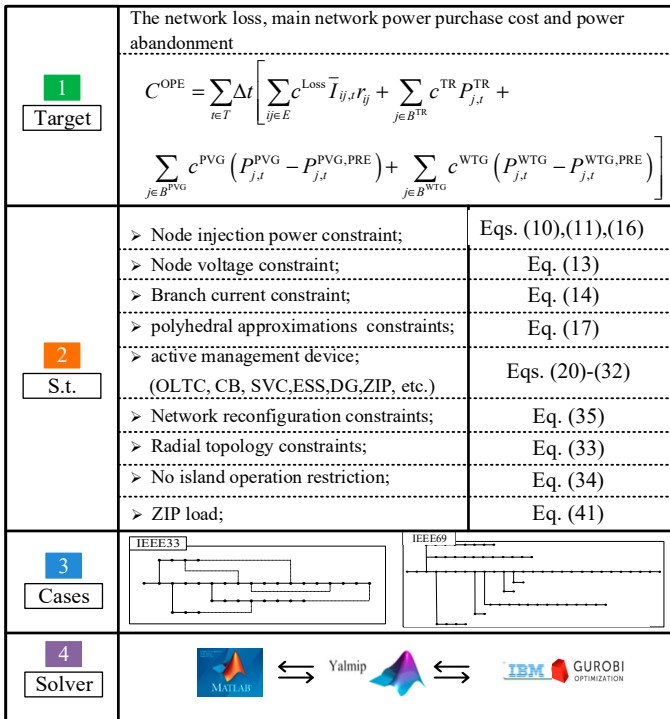

**Figure 3.** Proposed LSOCR-DOPF optimization flowchart.

## 3. Validation and Discussions

This paper mainly analyzes the application of three scenarios based on LSOCR-DOPF: (1) power coordination optimization; (2) network reconstruction; (3) ZIP load application. The numerical simulation analysis was carried out using an active distribution system based on the IEEE33 and IEEE69 bus systems data. The simulation parameters of the basic simulation system are presented in Appendix A. The simulation was performed on a 64-bit Windows 11 operating system with an Intel (R) Core (TM) i7-6700HQ CPU @ 2.64 GHz and 24 GB RAM, using MATLAB R2022b and calling YALMIP and GUROBI 10.0. The dynamic simulation period was set to 24 h. The voltage safety range (p.u.) was set to [0.94, 1.06].

### 3.1. Power Coordination Optimization

3.1.1. Scenario Description

This scenario (Figure 4) adds elements such as OLTC, ESS, CB, SVC, Wind, and photovoltaic (PV) to the basic IEEE33 network to comprehensively verify the application effect of each active management object in LSOCR-DOPF; assuming the OLTC transformation ratio range is 1–6%.

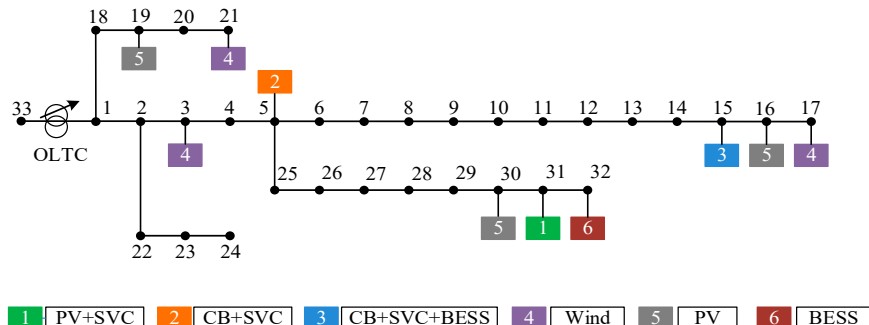

**Figure 4.** IEEE33 test active distribution network.

The validity of the similar relaxation model has been demonstrated in Refs. [14,17]. To further verify its universality, this scenario incorporates the cost of power purchase from the main network, as well as penalty costs for the abandonment of wind and photovoltaic power. The resulting total operational cost ($C^{\text{OPE}}$) is as follows:

$$C^{\text{OPE}} = \sum_{t \in T} \Delta t \left[ \sum_{ij \in E} c^{\text{Loss}} \bar{I}_{ij,t} r_{ij} + \sum_{j \in B^{\text{TR}}} c^{\text{TR}} P_{j,t}^{\text{TR}} + \sum_{j \in B^{\text{PVG}}} c^{\text{PVG}} \left( P_{j,t}^{\text{PVG}} - P_{j,t}^{\text{PVG,PRE}} \right) + \sum_{j \in B^{\text{WTG}}} c^{\text{WTG}} \left( P_{j,t}^{\text{WTG}} - P_{j,t}^{\text{WTG,PRE}} \right) \right] \tag{42}$$

where, $c^{\text{Loss}}$, $c^{\text{TR}}$, $c^{\text{WTG}}$, and $c^{\text{PVG}}$ are the network loss price, the main network power purchase price, the wind power abandonment price, and the photovoltaic power abandonment penalty price, respectively; $P_{j,t}^{\text{PVG}}$ and $P_{j,t}^{\text{WTG}}$ are the predicted output of wind power and photovoltaic, respectively.

3.1.2. Analysis of Optimization Results

The multi-period operational results of the power coordination and optimization of each active management unit are obtained based on the original data of the scenario, as depicted in Figures 5–10. Figures 5–9 show the active load demand, the active output of each wind turbine unit, the PV active output, the active output of the main grid, and the node injection power, respectively, as well as the active power of energy storage charging and discharging. Figure 10 presents the total 24-h CB reactive power output and SVC reactive power output.

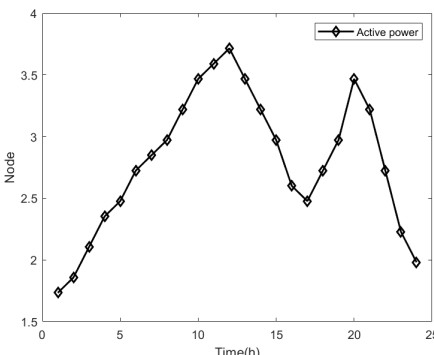

**Figure 5.** 24-h active load diagram.

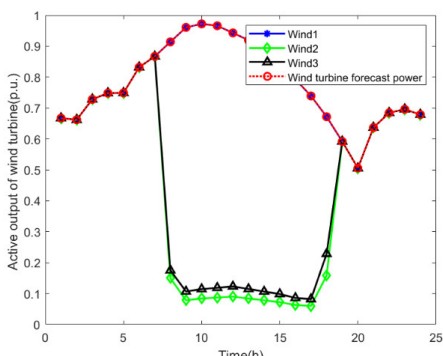

**Figure 6.** 24-h wind turbine output diagram.

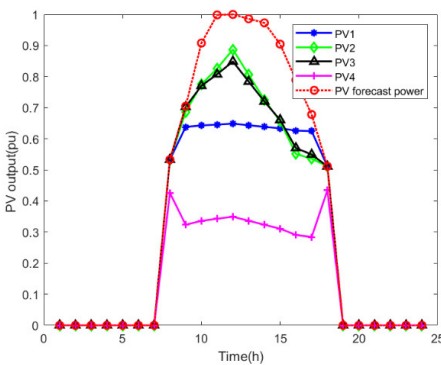

**Figure 7.** 24-h PV output diagram.

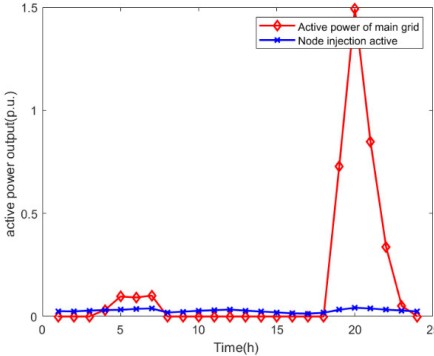

**Figure 8.** Active power output of main power grid and active power diagram of node injection in 24-h.

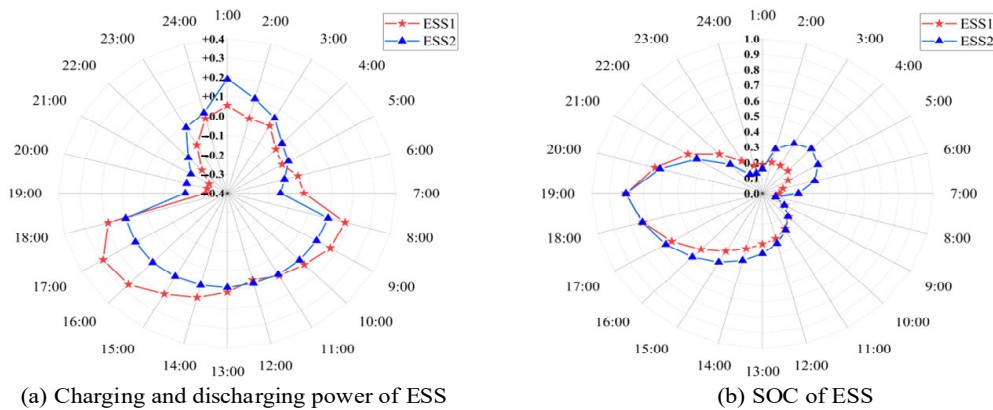

(a) Charging and discharging power of ESS

(b) SOC of ESS

**Figure 9.** SOC and charging and discharging of ESS.

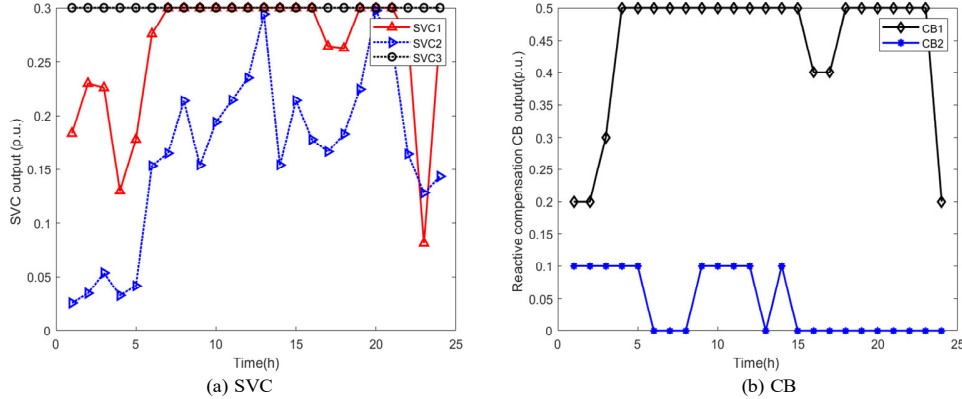

(a) SVC

(b) CB

**Figure 10.** 24-h reactive compensation power (SVC + CB) diagram.

It can be seen from Figures 5–10 that:

(1) During the peak load periods (08:00–15:00, 19:00–21:00), the active load demand of the system can be met while the Energy Storage System (ESS) is unable to absorb the surplus clean energy due to its own charging limitations, resulting in power abandonment. The ESS discharges during both the peak load period and the low peak period of renewable energy generation, effectively reducing the peak-valley difference of the equivalent load;

(2) During periods when the proportion of renewable energy output to load is relatively high (6:00–15:00, 18:00–22:00), reactive power compensation devices (Static Var Compensators (SVC) and Capacitor Banks (CB)) absorb the excess reactive power of the system, avoiding overvoltage.

Hence, the LSOCR-DOPF model proposed in this paper demonstrates significant effects on the optimization of both active and reactive power.

### 3.1.3. Model Validity Analysis

The LSOCR-DOPF is analyzed under three situations: "relaxation accuracy, calculation efficiency, and comparison of different optimization cases".

(1)  Relaxation accuracy

The relaxation accuracy of the objective function, such as network loss, has been proved and verified in the pure load network environment. However, considering the further analysis of the model accuracy after increasing the main network output, power abandonment, and load loss penalty costs, it is necessary to define the error index: $\Delta_{ij,t}^{\text{diff}} = \left| P_{ij,t}^2 + Q_{ij,t}^2 - \bar{I}_{i,t}\bar{U}_{j,t} \right|$.

Figure 11 shows the error scatter diagram of each branch in one day. Obviously, the deviation after relaxation meets the requirements of accurate operation, which is $10^{-6}$.

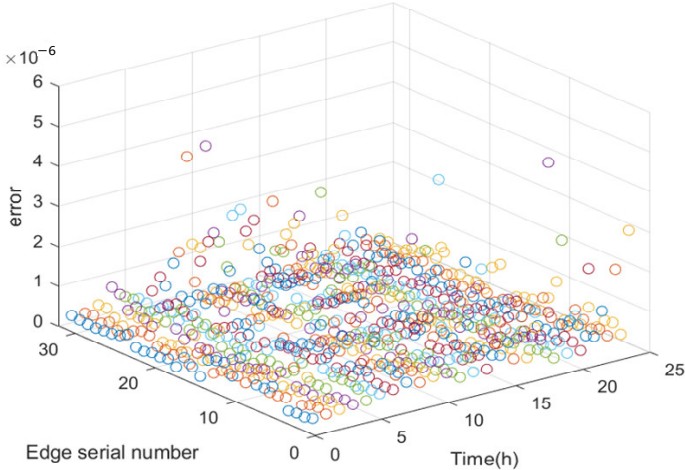

**Figure 11.** Scatter chart of error under each branch in each period.

(2) Calculation timeliness

Table 1 presents the solution speed and respective simulation results under various objective functions. Specifically, the objective functions of each simulated operating conditions are as follows: Case 1: minimized network loss; Case 2: minimized power purchase cost of the main grid; Case 3: minimum power loss; Case 4: minimized network loss and main grid power purchase cost; Case 5: minimized network loss, main grid power purchase cost, and power abandonment. Table 1 shows the solution speed and corresponding simulation results under different objective functions. Among them, the objective function Case 1 of each simulated operation condition is the minimum network loss; Case 2: the minimum power purchase cost of the main network; Case 3: minimal power loss; Case 4: network loss and main network power purchase cost are the least; Case 5: the network loss, main network power purchase cost, and power abandonment are the minimum.

**Table 1.** Optimization results under four types of objective functions.

| Case | Time (s) | Target ($10^3$ $) | | |
| --- | --- | --- | --- | --- |
| | | Network Loss | Power Purchase Cost | Power Abandonment |
| 1 | 20.394 | 0.244 | 10.937 | 22.939 |
| 2 | 18.581 | 3.274 | 1.890 | 10.728 |
| 3 | 8.555 | 10.013 | 4.757 | 6.752 |
| 4 | 125.044 | 0.337 | 1.890 | 13.673 |
| 5 | 200.482 | 4.796 | 1.890 | 9.087 |

Based on Table 1, with the increase in the objective function, the calculation time continues to increase, but the calculation speed is still acceptable, the maximum is 200 s, which meets the time requirements for day-ahead dispatching and real-time optimization of the active operation in the active distribution network.

Furthermore, based on Case 1, this part compares the solving efficiency of the MINLP (Interior Point-DOPF, Mixed-Integer Nonlinear Programming) model, the MISOCP (SOCR-DOPF) model, and the MILP (LSOCR-DOPF) model of the DOPF problem, and the results are summarized in Table 2. It can be seen from Table 2 that the accuracy and efficiency of LSOCR and SOCR are higher than those of the traditional interior point method. The distribution network loss optimization result of the MILP model linearized by SOCR is equal to the result of the MISOCP model, but the solution speed has been improved to a certain extent. This fully shows that the optimization efficiency can be improved by the LSOCR, which can reach 25~30% (compared with SOCR). It is shown that the LSOCR method is slightly better than the SOCR method in terms of comprehensive solution efficiency, but its solution timeliness is entirely better than the traditional interior point method.

**Table 2.** Comparison of optimization results of DOPF.

| Network | Target (10³ $) | | | Time (s) | | |
|---|---|---|---|---|---|---|
| | **MINLP** | **MISOCP** | **MILP** | **MINLP** | **MISOCP** | **MILP** |
| IEEE33 | 0.352 | 0.244 | 0.244 | >5 h | 20.394 | 14.564 |
| IEEE69 | 1.556 | 1.356 | 1.356 | >5 h | 27.396 | 20.509 |

(3)    Comparison of different optimization cases

According to Tables 1 and 2, it can be seen that:

① In Case 1, only the optimization of network loss is considered, leading to substantial waste of clean energy power. On the other hand, Case 2 prioritizes the minimization of main grid power purchase costs, which results in increased consumption of clean energy, reducing the main grid power purchase and, in turn, reducing power waste. Conversely, Case 3 focuses on minimizing power loss. In comparison to Case 1, it requires more utilization of clean energy to reduce main grid power purchases, but results in an increase in network loss;

② In Case 4, the objective is to simultaneously minimize both the network loss and the main network's power purchase. The results show that while the main network's power purchase remains unchanged, the network loss decreases, but the amount of power abandonment increases. The paper assumes that the cost of network loss and the penalty cost of power abandonment are both equal to 5000 $/MWh, which suggests that excessive access to clean energy may increase the network loss of the system;

③ Case 5 aims to minimize the network loss, main network power purchase, and power abandonment. The main network power purchase remains the same, while the power abandonment correspondingly decreases, and the network loss increases, which means that the network loss caused by increasing the access to clean energy is less than the penalty of clean energy power abandonment, so the power abandonment is further reduced.

④ To assess the impact of the active management equipment (SVC, CB, ESS) introduced in this paper on the performance of the distribution network, three tests were designed and included: (a) No addition of energy storage and reactive power compensation equipment; (b) Addition of an energy storage device in place of a reactive compensation device; (c) Simultaneous addition of both an energy storage device and a reactive compensation device. It can be seen from Figure 12 that adding the active management unit can improve the uniform distribution of voltage in the distribution network to some extent, which shows some functions of the active management unit. It is worth mentioning that the relatively elevated node voltage depicted in Figure 12 can be attributed to the network loss specified by the objective function.

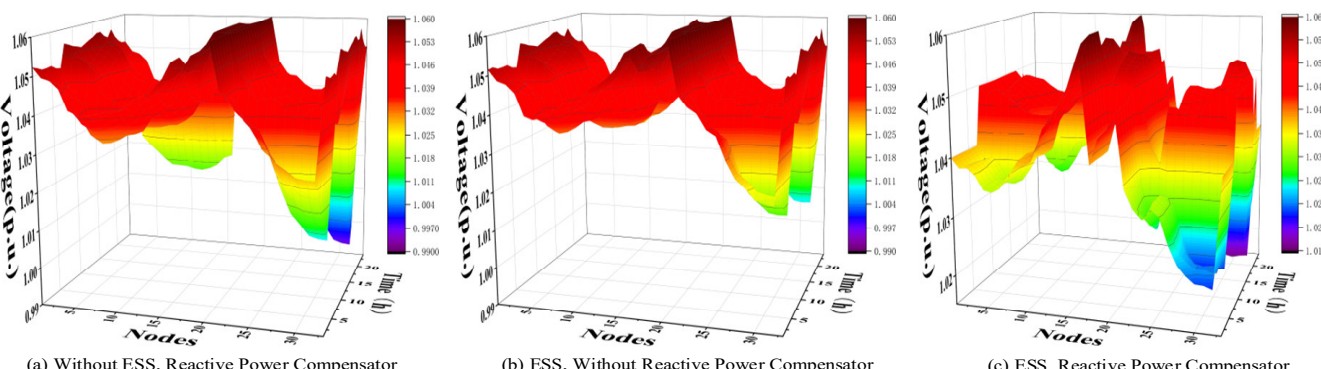

(a) Without ESS, Reactive Power Compensator    (b) ESS, Without Reactive Power Compensator    (c) ESS, Reactive Power Compensator

**Figure 12.** Voltage of different nodes (comparative verification).

Based on the simulation results of the scenario in this section, an analysis of three important parameters, calculation time, solution error, and node voltage distribution, is conducted to evaluate the effectiveness of the proposed method in this paper.

Firstly, in terms of power grid stability, as shown in Figure 12c, the voltage distribution of each node in the IEEE33 node network can meet the set range requirement of [0.94, 1.06], and the distribution is relatively uniform.

Secondly, in terms of power grid scheduling frequency, as shown in Tables 1 and 2, the six sets of comparison cases (IEEE33 + IEEE69) set in this scenario can run within 3 min (minimum of 8.555 s), which can meet the real-time power grid scheduling frequency requirements.

Thirdly, in terms of solving accuracy, as shown in Figure 11, the magnitude of the error in the simulation results of this scenario is strictly controlled within 10-6, which can ensure the requirement of power grid solving accuracy.

### 3.2. Network Reconfiguration

The distribution network reconfiguration scenario is analyzed based on the standard IEEE 33 systems (Appendix A), and the load value is the original system data. In order to verify the effectiveness of second-order cone relaxation in distribution network reconstruction, this example uses static single-period reconstruction for analysis. In the two examples, node 1 is a substation node with a voltage amplitude of 1.06 p.u.. It is assumed that each branch is equipped with a section switch, and the optimization calculation is carried out with the goal of minimizing the network loss. The results are shown in Table 3. This also solves the error scatter diagram of each branch, as shown in Figure 13.

**Table 3.** Reconfiguration scheme and optimization results.

| IEEE33 | Time (s) | Network Loss (MW) | Original Network Loss (MW) | Disconnected Switch |
|---|---|---|---|---|
| Single-period | 1.355014 | 0.0256 | 0.0368 | 7 (6–7), 9 (8–9), 14 (13–14), 32 |
| Multi-period | 153.404963 | 1.7080 | 2.4964 | (31–32), 37 (24–28) |

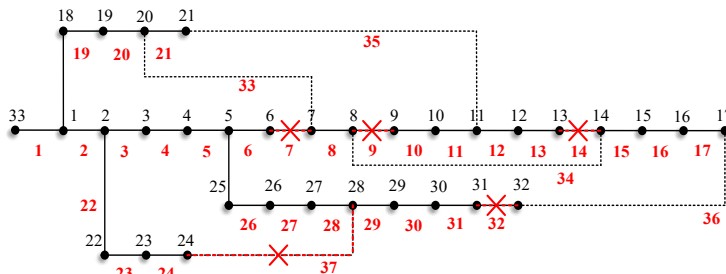

**Figure 13.** Optimized and reconstructed IEEE33 network (node serial number is given by black numbers, branch serial number is shown by red numbers, and red cross and red dotted line indicate disconnected branch).

As shown in Figure 14, it is not difficult to find that the error is $10^{-9}$ orders of magnitude, meeting the reconstruction requirements. It is worth noting that, in this example, no active management equipment is added. For IEEE 33 nodes, all nodes have load values, so only Equation (33) can be added to the radiation constraint. For some nodes in the IEEE69 system without load, Equation (34) must be added to ensure that all nodes are connected and operate without islands and radiation.

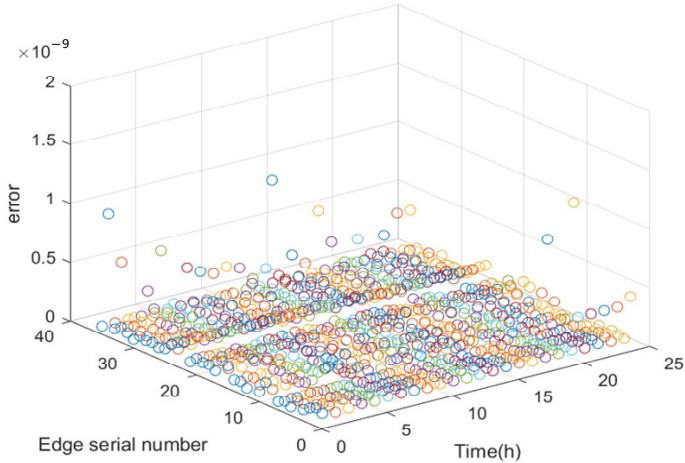

**Figure 14.** Scatter diagram of distribution network reconfiguration error.

Based on the simulation results of this section, the three important parameters of calculation time, solution error, and node voltage distribution are evaluated. Firstly, in terms of power grid stability, as shown in Figure 15, the voltage distribution of each node in the IEEE33 node network can meet the set range requirement of [0.94, 1.06], and the voltage distribution in multi-period is relatively uniform. Secondly, regarding power grid scheduling frequency, as shown in Table 3, the two sets of comparative cases (single-period and multi-period) ensure that the operating time is within 3 min (the shortest being 1.355 s), meeting the requirements of real-time power grid scheduling frequency. Thirdly, in terms of solving accuracy, as shown in Figure 14, the error level of the simulation results in this scenario is strictly controlled within $10^{-9}$, which can well guarantee the accuracy requirements of power grid solving.

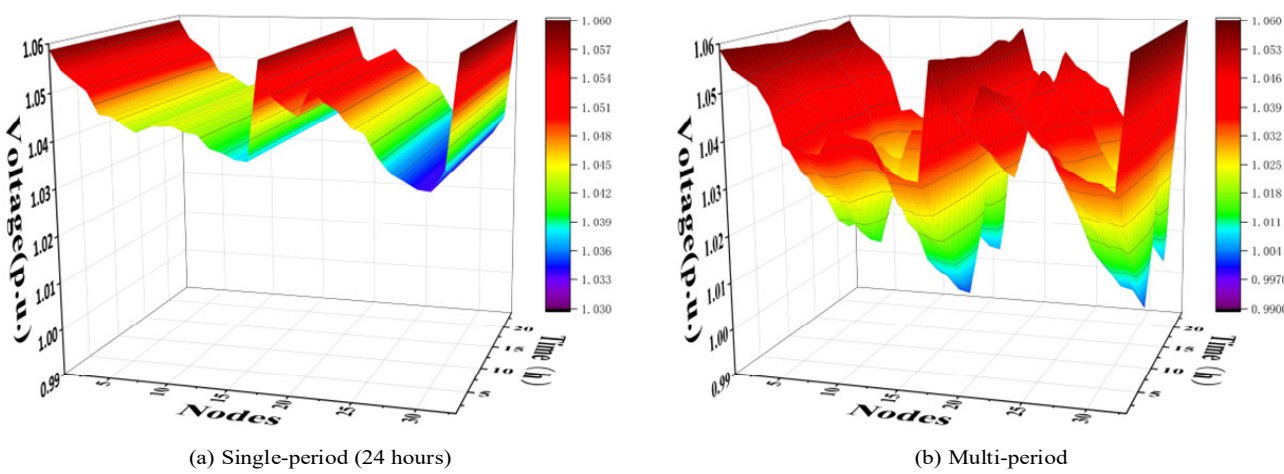

(a) Single-period (24 hours)　　　　　　　　　　　(b) Multi-period

**Figure 15.** Voltage of different nodes.

### 3.3. ZIP Load Application

This scenario adopts the PG69 node system test (Figure 16). In order test the static voltage characteristics of the load, this part does not consider the active management unit, and only analyzes the calculation results after citing the static voltage characteristics of the load. Three types of loads, namely constant power, constant current, and constant impedance, are added to each node. The voltage distribution of each node at each time is calculated as shown in Figure 17. Figure 18 shows the comparison of load active demand considering voltage static characteristics or not.

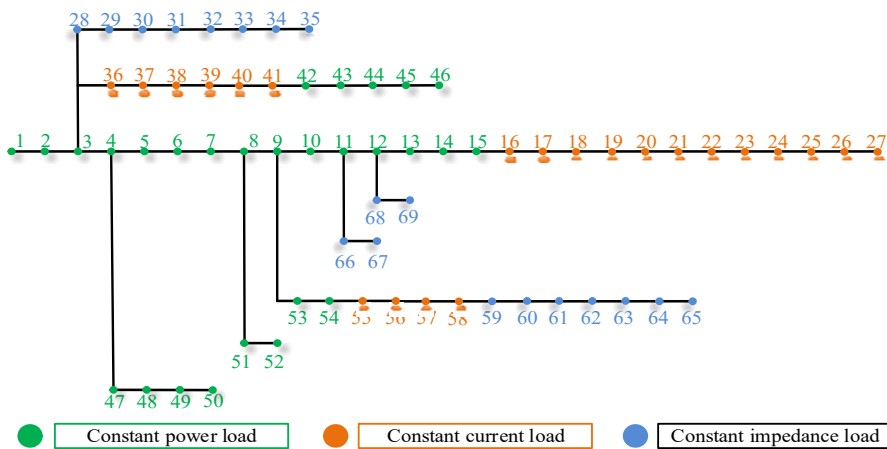

**Figure 16.** IEEE69 test active distribution network.

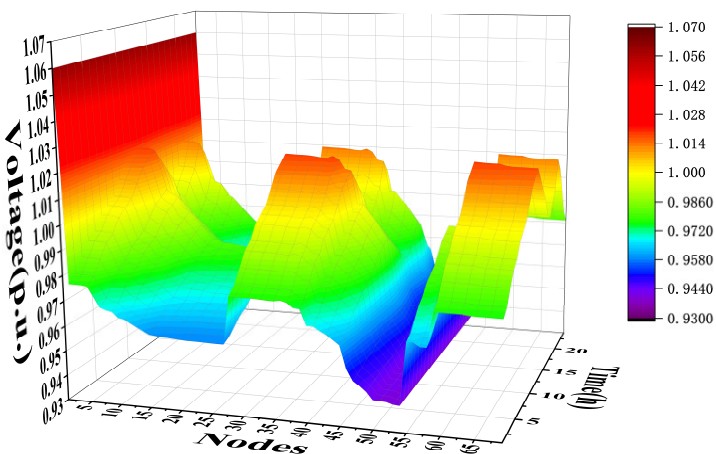

**Figure 17.** Voltage distribution diagram of each node.

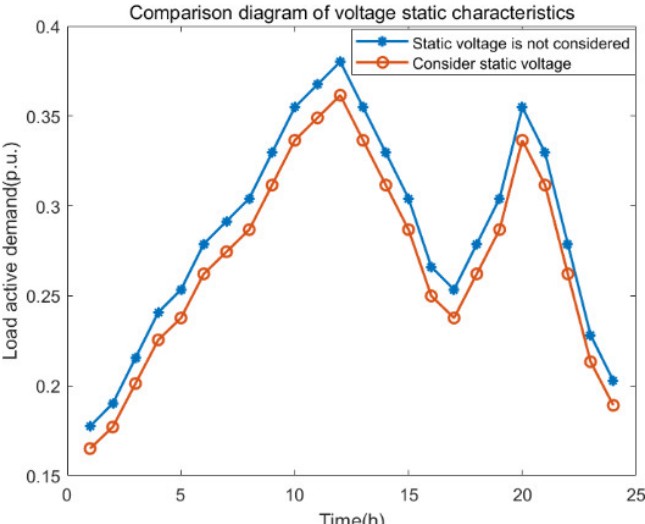

**Figure 18.** Comparison of load active demand considering voltage static characteristics or not.

As can be seen from Figures 17 and 18, since most of the node voltage is lower than 1.0 p.u., the load demand will be reduced after considering the static voltage characteristics of the load. It is important to consider the static voltage characteristics of active distribution

network fine simulation. In addition, the second-order cone relaxation error of each period is shown in Figure 19. Obviously, the relaxation effect is also highly satisfactory.

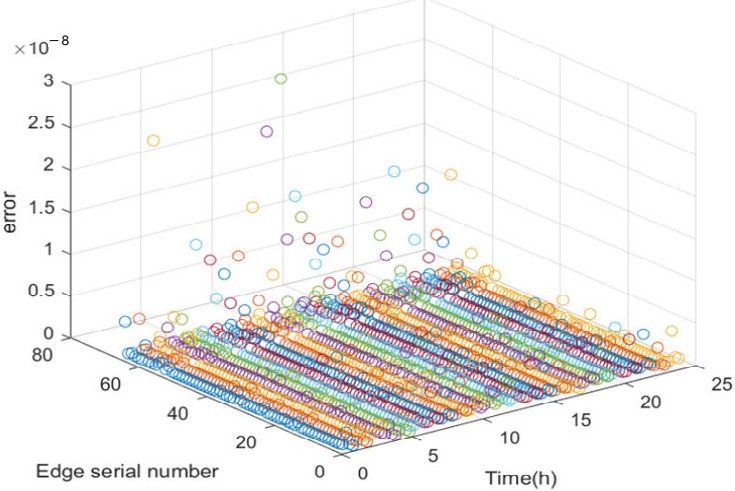

**Figure 19.** Error scatter chart of ZIP load model.

Based on the simulation results of this scenario, the calculation time, solution error, and node voltage distribution are evaluated. Firstly, in terms of power grid stability, as shown in Figure 17, the voltage distribution of each node in the IEEE33 node network can meet the requirements of the [0.94, 1.06] range. Secondly, in terms of power grid dispatch frequency, the running time of this case is 5.394 s, which can meet the requirements of real-time power grid dispatch frequency. Thirdly, in terms of solution accuracy, as shown in Figure 19, the error level of the simulation result in this scenario is strictly controlled within $10^{-8}$, which can guarantee the requirement of power grid solution accuracy. Finally, regarding the accurate modeling of distribution networks, as shown in Figure 18, the introduction of ZIP loads can change the load demand, demonstrating the importance of the active and detailed modeling of distribution networks.

## 4. Conclusions

In this paper, a polyhedral linear approximation method of the second-order cone relaxation (LSOCR-DOPF) for the dynamic optimal power flow problem based on the branch power flow model of the distribution network is proposed and validated. Detailed linear constraint modeling of active management units is performed, and the effectiveness of the proposed solution method is verified through comparison with other solutions in three major application scenarios. The results show that the strategy proposed in this paper can meet the engineering standards for typical application scenarios of distribution networks: on one hand, the calculation time is controlled within 3 min, which satisfies the requirement of the general power grid scheduling refresh rate and ensures fast solution of OPF problems in future large-scale active distribution networks; on the other hand, the calculation result error is controlled within $10^{-6}$, effectively avoiding the catastrophic consequences caused by calculation errors in previous work; moreover, the voltage distribution of each node in the distribution network meets the condition constraints [0.94, 1.06] set by authors, which can achieve the optimization goal of reducing network loss. It is noteworthy that the LSOCR-DOPF model satisfies the requirements of calculation time and accuracy in the fields of daily scheduling, real-time operation, and the control of active distribution networks, demonstrating strong practical application value, and the case study suggests that considering the ZIP model for the fine simulation of active distribution networks is also of great significance. Currently, the inclusion of discrete variables, which are non-convex sources, has an impact on the accuracy of the relaxation model, and the universality condi-

tion of the second-order cone relaxation model requires further theoretical investigation in future work.

Furthermore, it is difficult to ensure the accuracy of convex relaxation using a relatively universal method due to the various boundary conditions that need to be set in the model for practical applications. The technical challenges of applying convex relaxation techniques in engineering can be summarized as follows: further exploring the influence of the objective function and feasible region of the OPF problem on convex relaxation and seeking sufficient conditions to guarantee accurate convex relaxation in theory; constructing tighter and more precise relaxations based on SOCP relaxation techniques and exploring the possibility of combining convex relaxation techniques with other OPF solution methods.

**Author Contributions:** Conceptualization, W.M. and X.D.; methodology, W.M.; software, W.M.; validation, W.M. and J.Y.; formal analysis, W.M. and R.M.R.-A.; investigation, M.D.; resources, W.M.; data curation, W.M. and X.D.; writing—original draft preparation, X.D. and W.M.; writing—review and editing, V.S.; visualization, W.M.; supervision, D.S.; project administration, X.D.; funding acquisition, X.D. All authors have read and agreed to the published version of the manuscript.

**Funding:** This research was supported by National Natural Science Foundation of China (NSFC) under Grant 52177204; the Natural Science Foundation of Hunan Province (No. 2020JJ4744); the Innovation-Driven Project of Central South University (No.2020CX031); the internal grant project of VSB-Technical University of Ostrava (SGS project, grant number SP2022/77).

**Data Availability Statement:** Not applicable.

**Conflicts of Interest:** The authors declare no conflict of interest.

**Appendix A**

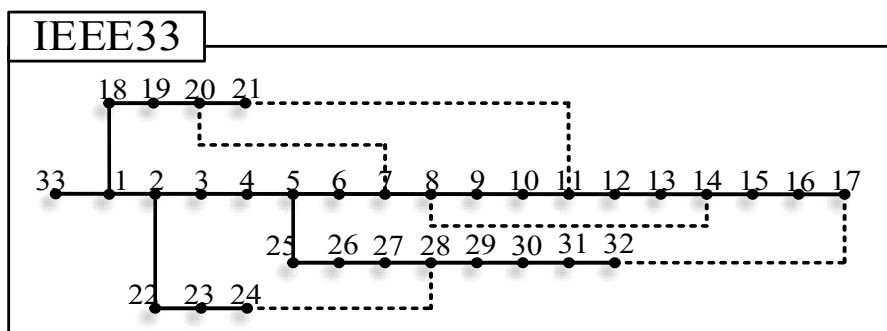

**Figure A1.** IEEE33 Radial Distribution Network.

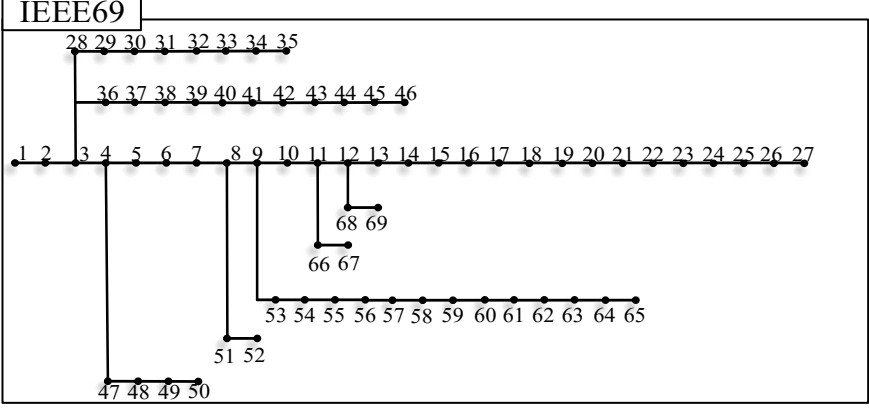

**Figure A2.** IEEE69 Radial Distribution Network.

**Table A1.** System data for 69-bus radial distribution network ('*' denotes a tie-line).

| Branch Number | Sending Bus | Receiving Bus | Resistance Ω | Reactance Ω | Nominal Load at Receiving Bus | | Maximum Line Capacity (kVA) |
| --- | --- | --- | --- | --- | --- | --- | --- |
| | | | | | P (kW) | Q(kVA) | |
| 1 | 1 | 2 | 0.0005 | 0.0012 | 0.0 | 0.0 | 10,761 |
| 2 | 2 | 3 | 0.0005 | 0.0012 | 0.0 | 0.0 | 10,761 |
| 3 | 3 | 4 | 0.0015 | 0.0036 | 0.0 | 0.0 | 10,761 |
| 4 | 4 | 5 | 0.0251 | 0.0294 | 0.0 | 0.0 | 5823 |
| 5 | 5 | 6 | 0.3660 | 0.1864 | 2.60 | 2.20 | 1899 |
| 6 | 6 | 7 | 0.3811 | 0.1941 | 40.40 | 30.00 | 1899 |
| 7 | 7 | 8 | 0.0922 | 0.0470 | 75.00 | 54.00 | 1899 |
| 8 | 8 | 9 | 0.0493 | 0.0251 | 30.00 | 22.00 | 1899 |
| 9 | 9 | 10 | 0.8190 | 0.2707 | 28.00 | 19.00 | 1455 |
| 10 | 10 | 11 | 0.1872 | 0.0619 | 145.00 | 104.00 | 1455 |
| 11 | 11 | 12 | 0.7114 | 0.2351 | 145.00 | 104.00 | 1455 |
| 12 | 12 | 13 | 1.0300 | 0.3400 | 8.00 | 5.00 | 1455 |
| 13 | 13 | 14 | 1.0440 | 0.3450 | 8.00 | 5.50 | 1455 |
| 14 | 14 | 15 | 1.0580 | 0.3496 | 0.0 | 0.0 | 1455 |
| 15 | 15 | 16 | 0.1966 | 0.0650 | 45.50 | 30.00 | 1455 |
| 16 | 16 | 17 | 0.3744 | 0.1238 | 60.00 | 35.00 | 1455 |
| 17 | 17 | 18 | 0.0047 | 0.0016 | 60.00 | 35.00 | 2200 |
| 18 | 18 | 19 | 0.3276 | 0.1083 | 0.0 | 0.0 | 1455 |
| 19 | 19 | 20 | 0.2106 | 0.0690 | 1.00 | 0.60 | 1455 |
| 20 | 20 | 21 | 0.3416 | 0.1129 | 114.00 | 81.00 | 1455 |
| 21 | 21 | 22 | 0.0140 | 0.0046 | 5.00 | 3.50 | 1455 |
| 22 | 22 | 23 | 0.1591 | 0.0526 | 0.0 | 0.0 | 1455 |
| 23 | 23 | 24 | 0.3463 | 0.1145 | 28.00 | 20.0 | 1455 |
| 24 | 24 | 25 | 0.7488 | 0.2475 | 0.0 | 0.0 | 1455 |
| 25 | 25 | 26 | 0.3089 | 0.1021 | 14.0 | 10.0 | 1455 |
| 26 | 26 | 27 | 0.1732 | 0.0572 | 14.0 | 10.0 | 1455 |
| 27 | 3 | 28 | 0.0044 | 0.0108 | 26.0 | 18.6 | 10,761 |
| 28 | 28 | 29 | 0.0640 | 0.1565 | 26.0 | 18.6 | 10,761 |
| 29 | 29 | 30 | 0.3978 | 0.1315 | 0.0 | 0.0 | 1455 |
| 30 | 30 | 31 | 0.0702 | 0.0232 | 0.0 | 0.0 | 1455 |
| 31 | 31 | 32 | 0.3510 | 0.1160 | 0.0 | 0.0 | 1455 |
| 32 | 32 | 33 | 0.8390 | 0.2816 | 14.0 | 10.0 | 2200 |
| 33 | 33 | 34 | 1.7080 | 0.5646 | 9.50 | 14.00 | 1455 |
| 34 | 34 | 35 | 1.4740 | 0.4873 | 6.00 | 4.00 | 1455 |
| 35 | 3 | 36 | 0.0044 | 0.0108 | 26.0 | 18.55 | 10,761 |
| 36 | 36 | 37 | 0.0640 | 0.1565 | 26.0 | 18.55 | 10,761 |
| 37 | 37 | 38 | 0.1053 | 0.1230 | 0.0 | 0.0 | 5823 |
| 38 | 38 | 39 | 0.0304 | 0.0355 | 24.0 | 17.00 | 5823 |
| 39 | 39 | 40 | 0.0018 | 0.0021 | 24.0 | 17.00 | 5823 |
| 40 | 40 | 41 | 0.7283 | 0.8509 | 1.20 | 1.0 | 5823 |

**Table A1.** *Cont.*

| Branch Number | Sending Bus | Receiving Bus | Resistance Ω | Reactance Ω | Nominal Load at Receiving Bus | | Maximum Line Capacity (kVA) |
|---|---|---|---|---|---|---|---|
| | | | | | P (kW) | Q(kVA) | |
| 41 | 41 | 42 | 0.3100 | 0.3623 | 0.0 | 0.0 | 5823 |
| 42 | 42 | 43 | 0.0410 | 0.0478 | 6.0 | 4.30 | 5823 |
| 43 | 43 | 44 | 0.0092 | 0.0116 | 0.0 | 0.0 | 5823 |
| 44 | 44 | 45 | 0.1089 | 0.1373 | 39.22 | 26.30 | 5823 |
| 45 | 45 | 46 | 0.0009 | 0.0012 | 39.22 | 26.30 | 6709 |
| 46 | 4 | 47 | 0.0034 | 0.0084 | 0.00 | 0.0 | 10,761 |
| 47 | 47 | 48 | 0.0851 | 0.2083 | 79.00 | 56.40 | 10,761 |
| 48 | 48 | 49 | 0.2898 | 0.7091 | 384.70 | 274.50 | 10,761 |
| 49 | 49 | 50 | 0.0822 | 0.2011 | 384.70 | 274.50 | 10,761 |
| 50 | 8 | 51 | 0.0928 | 0.0473 | 40.50 | 28.30 | 1899 |
| 51 | 51 | 52 | 0.3319 | 0.1114 | 3.60 | 2.70 | 2200 |
| 52 | 52 | 53 | 0.1740 | 0.0886 | 4.35 | 3.50 | 1899 |
| 53 | 53 | 54 | 0.2030 | 0.1034 | 26.40 | 19.00 | 1899 |
| 54 | 54 | 55 | 0.2842 | 0.1447 | 24.00 | 17.20 | 1899 |
| 55 | 55 | 56 | 0.2813 | 0.1433 | 0.0 | 0.0 | 1899 |
| 56 | 56 | 57 | 1.5900 | 0.5337 | 0.0 | 0.0 | 2200 |
| 57 | 57 | 58 | 0.7837 | 0.2630 | 0.0 | 0.0 | 2200 |
| 58 | 58 | 59 | 0.3042 | 0.1006 | 100.0 | 72.0 | 1455 |
| 59 | 59 | 60 | 0.3861 | 0.1172 | 0.0 | 0.0 | 1455 |
| 60 | 60 | 61 | 0.5075 | 0.2585 | 1244.0 | 888.00 | 1899 |
| 61 | 61 | 62 | 0.0974 | 0.0496 | 32.0 | 23.00 | 1899 |
| 62 | 62 | 63 | 0.1450 | 0.0738 | 0.0 | 0.0 | 1899 |
| 63 | 63 | 64 | 0.7105 | 0.3619 | 227.0 | 162.00 | 1899 |
| 64 | 64 | 65 | 1.0410 | 0.5302 | 59.0 | 42.0 | 1899 |
| 65 | 11 | 66 | 0.2012 | 0.0611 | 18.0 | 13.0 | 1455 |
| 66 | 66 | 67 | 0.0047 | 0.0014 | 18.0 | 13.0 | 1455 |
| 67 | 12 | 68 | 0.7394 | 0.2444 | 28.0 | 20.0 | 1455 |
| 68 | 68 | 69 | 0.0047 | 0.0016 | 28.0 | 20.0 | 1455 |
| 69 * | 11 | 43 | 0.5000 | 0.5000 | | | 566 |
| 70 * | 13 | 21 | 0.5 | 0.5 | | | 566 |
| 71 * | 15 | 46 | 1.0 | 1.0 | | | 400 |
| 72 * | 50 | 59 | 2.0 | 2.0 | | | 283 |
| 73 * | 27 | 65 | 1.0 | 1.0 | | | 400 |

**Table A2.** System data for 33-bus radial distribution network.

| Branch Number | Sending Bus | Receiving Bus | Resistance Ω | Reactance Ω | Nominal Load at Receiving Bus | |
|---|---|---|---|---|---|---|
| | | | | | P (kW) | Q (kVA) |
| 1 | 1 | 2 | 0.0922 | 0.047 | 100 | 60 |
| 2 | 2 | 3 | 0.493 | 0.2511 | 90 | 40 |
| 3 | 3 | 4 | 0.366 | 0.1864 | 120 | 80 |
| 4 | 4 | 5 | 0.3811 | 0.1941 | 60 | 30 |
| 5 | 5 | 6 | 0.819 | 0.707 | 60 | 20 |
| 6 | 6 | 7 | 0.1872 | 0.6188 | 200 | 100 |
| 7 | 7 | 8 | 0.7114 | 0.2351 | 200 | 100 |
| 8 | 8 | 9 | 1.03 | 0.74 | 60 | 20 |
| 9 | 9 | 10 | 1.044 | 0.74 | 60 | 20 |
| 10 | 10 | 11 | 0.1966 | 0.065 | 45 | 30 |
| 11 | 11 | 12 | 0.3744 | 0.1298 | 60 | 35 |
| 12 | 12 | 13 | 1.468 | 1.155 | 60 | 35 |
| 13 | 13 | 14 | 0.5416 | 0.7129 | 120 | 80 |
| 14 | 14 | 15 | 0.591 | 0.526 | 60 | 10 |
| 15 | 15 | 16 | 0.7463 | 0.545 | 60 | 20 |
| 16 | 16 | 17 | 1.289 | 1.721 | 60 | 20 |
| 17 | 17 | 18 | 0.732 | 0.574 | 90 | 40 |
| 18 | 2 | 19 | 0.164 | 0.1565 | 90 | 40 |
| 19 | 19 | 20 | 1.5042 | 1.3554 | 90 | 40 |
| 20 | 20 | 21 | 0.4095 | 0.4784 | 90 | 40 |
| 21 | 21 | 22 | 0.7089 | 0.9373 | 90 | 40 |
| 22 | 3 | 23 | 0.4512 | 0.3083 | 90 | 50 |
| 23 | 23 | 24 | 0.898 | 0.7091 | 420 | 200 |
| 24 | 24 | 25 | 0.896 | 0.7011 | 420 | 200 |
| 25 | 6 | 26 | 0.203 | 0.1034 | 60 | 25 |
| 26 | 26 | 27 | 0.2842 | 0.1447 | 60 | 25 |
| 27 | 27 | 28 | 1.059 | 0.9337 | 60 | 20 |
| 28 | 28 | 29 | 0.8042 | 0.7006 | 120 | 70 |
| 29 | 29 | 30 | 0.5075 | 0.2585 | 200 | 600 |
| 30 | 30 | 31 | 0.9744 | 0.963 | 150 | 70 |
| 31 | 31 | 32 | 0.3105 | 0.3619 | 210 | 100 |
| 32 | 32 | 33 | 0.341 | 0.5302 | 60 | 40 |
| 33 | 20 | 7 | 2.0000 | 2.0000 | - | - |
| 34 | 8 | 14 | 2.0000 | 2.0000 | - | - |
| 35 | 11 | 21 | 2.0000 | 2.0000 | - | - |
| 36 | 17 | 32 | 0.5000 | 0.5000 | - | - |
| 37 | 24 | 28 | 0.5000 | 0.5000 | - | - |

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
