# Peer review of "Dynamic Optimal Power Flow of Active Distribution Network Based on LSOCR and Its Application Scenarios"

_electronics, doi:10.3390/electronics12071530_

Round 1
Reviewer 1 Report
1. add open challenges in a similar domain for future researchers.
2. Significance of the equations should be appropriately mentioned.
3. justify why linearized second-order conic relaxation?
4. few more evaluation parameters should be added for more justified results.
Author Response
Responses to Reviewer 1
(Notes: In the revised manuscript, the main changes are highlighted in yellow for clarity and easy reference.)
Comment 1: add open challenges in a similar domain for future researchers.
|
Response 1: Thanks for your comment on our paper. In order to provide research references for scholars in relevant fields, we have added two open challenges in a similar domain for future researchers in the conclusion. Revised manuscript: Currently, the inclusion of discrete variables, which are non-convex sources, has an impact on the accuracy of the relaxation model, and the universality condition of the second-order cone relaxation model requires further theoretical investigation in future work. Further, it is difficult to ensure the accuracy of convex relaxation using a relatively universal method due to the various boundary conditions that need to be set in the model for practical applications. The technical challenges of applying convex relaxation techniques in engineering can be summarized as follows: further exploring the influence of the objective function and feasible region of the OPF problem on convex relaxation and seeking sufficient conditions to guarantee accurate convex relaxation in theory; constructing tighter and more precise relaxations based on SOCP relaxation techniques and exploring the possibility of combining convex relaxation techniques with other OPF solution methods. (in the last paragraph of the conclusion) |
Comment 2: Significance of the equations should be appropriately mentioned.
|
Response 2: We are very grateful for the kind advice of the dear reviewer. As suggested, we highlighted the important equations in the modeling process with a flowchart in Figure 2 and explained them in detail in the context, so that readers can intuitively understand the key modeling equations. In addition, we described in Figure 3 the linear constraints of the active management units incorporated into the LSOCR-DOPF model, to provide a comprehensive understanding of the model. All the corrections are marked in the revised manuscript. Revised manuscript: …… Besides, and represents complex power, and . And branch complex impedance ; Set represents the set of all nodes in the network; In the traditional OPF, the voltage remains constant. If OLTC is installed, the voltage will change with the OLTC transformation ratio; E represents the collection of all branches in the network. There are substations, nodes and line branches in the network.
Figure 1. Structure of radial distribution network. 2.1.2 Basic OPF model based on BFM Generally, the basic model of optimal power flow based on branch power flow (BFM-OPF, Figure 1) is expressed as follows [14]: (1) s.t. (2) (3) (4) (5) (6) (7) where, and represent the active and reactive power injections, respectively, at each node; the branch represents the positive direction of flow direction from node to node ; is the collection of branch end nodes with as the head node, and is the collection of branch end nodes with as the end node; and denote the active and reactive power at the head node of the branch , respectively; and denote the active and reactive current flow in each branch; and correspond to the resistance and reactance of each branch, individually; and are the ground conductance and ground susceptance of node , separately; and are the upper and lower limits of node voltage, respectively; and are the upper and lower limits of branch current, respectively. From equations (1) to (7), it can be inferred that: 1) The optimization variables for OPF consist of node injection power , branch power flow , node voltage , and branch current , with the substation node voltage not considered as an optimization variable; 2) Equation (1) represents the objective function, which can be the minimization of network losses and substation node power purchases; 3) The well-known branch flow equation [30] is expressed by equations (2)-(3), while equation (4) represents the power calculation equation. Equations (5) and (6) respectively denote the safety constraint equations for branch currents and nodal voltages. Equation (7) introduces node-dependent constraints that are subject to change based on the employed model. …… (8) Substitute equation (8) into equations (2) ~ (6) to change the power flow constraint to the second-order cone constraint [31] as follows: …… With this, the SOCR-OPF (a typical Mixed-Integer Second Order Cone Programming, MISOCP) model is fully modeled. Among them, the second-order cone programming represented by equation (12) can be expanded and simplified as : (15) The equation (15) can be uniformly described as: (16) Ben-Tal and Nemirovski [22] show that equation (16) can be approximated by a system of linear homogeneous equalities and inequalities in terms of , and variables for . is a parameter of the polyhedral-relaxed approximation. The approximate expression of the polyhedron of the three-dimensional SOCR constraint (equation (16)) is: (17) where, represents the number of facets in a polyhedron. and are auxiliary variables. The polyhedral approximation given by (17) can be reduced by using the linear equality constraints to solve for in terms of and and then substitute out of the equation (17). The resulting system will only have linear inequality constraints in terms of , and the variables for . The error ( ) [22] of the polyhedral approximations of the three-dimensional second-order cone constraint (Equation (16)) is: (18) According to Equation (18), when K=11, the error is about . Therefore, the MISOCP model for active distribution network reconfiguration is approximately equivalent to the MILP model. At this point, the improved version of SOCR-OPF, namely the LSOCR-OPF model, is fully modeled. So far, the power flow constraint has been transformed from a nonlinear programming model to a MISOCP model composed of equations (9) ~ (14). Then, through the polyhedron approximation method which is used to linearize equation (16), the MISOCP model can be approximately converted into a MILP model for solution. ……
Figure 2. The Proposed modeling process of LSOCR for solving AC-OPF. (on the lines 267-268 of page 7) ……
Figure 3. Proposed LSOCR-DOPF optimization flowchart. (on the lines 430-431 of page 12) ……
|
Comment 3: justify why linearized second-order conic relaxation?
|
Response 3: Thanks for your suggestion. Due to the authors' negligence, in the original manuscript, the reasons of using linearized second-order conic relaxation were not well given in detail in the introduction. In the revised manuscript, we explain in detail why we use the linearized second-order cone relaxation. Revised manuscript: Additionally, Refs. [21] conducted a thorough and precise approximation proof of the polyhedral solid approximation method in second-order cone programming, enabling the direct linearization of equivalent SOCR and improving the efficiency of solution models while maintaining optimal solution quality. although the OPF model of the active distribution network based on MISOCP has achieved a relatively high solution efficiency [21], strictly speaking, it is still a non-linear model, and the efficiency of solving nonlinear second-order cone constraints will decrease with the increase of the number of distribution network nodes. In order to achieve efficient solving of the OPF problem in future large-scale distribution networks, it is necessary to simplify the second-order cone programming constraints while ensuring the accuracy and efficiency of distribution network solving. Ref. [22] has proven that the second-order cone in dimension N, can be outer-approximated to an arbitrary accuracy by a polyhedral cone in an extended space. Polyhedral approximations are very powerful for solving MISOCP as the benefits of warm-starting for LPs can be utilized throughout the branch-and-bound algorithm [23]. This is particularly important for problems that can be formulated as MISOCP, such as the OPF problem in this paper. Meanwhile, Refs. [24,25] extended the constant power load model to a ZIP load model under a rectangular coordinate system and applied it to the solution of the OPF. (on the lines 102~115 of page 3)
|
Comment 4: few more evaluation parameters should be added for more justified results.
|
Response 4: Thanks for your comment on our paper. To follow your valuable suggestion, we separately supplemented a concluding paragraph in each of the three scenarios to explain the most important technical parameters used to evaluate the distribution network: solution error (representing the model's accuracy), calculation time (related to power grid scheduling refresh rate, representing the dynamic characteristics of the grid), and node voltage distribution (representing stable operation of the power grid). It is given in the following: Revised manuscript: 3.1 Power coordination optimization …… Figure 12. Voltage of different nodes (comparative verification). Based on the simulation results of the scenario in this section, an analysis of three important parameters, calculation time, solution error, and node voltage distribution, is conducted to evaluate the effectiveness of the proposed method in this paper. Firstly, in terms of power grid stability, as shown in Figure 12(c), the voltage distribution of each node in the IEEE33 node network can meet the set range requirement of [0.94,1.06], and the distribution is relatively uniform. Secondly, in terms of power grid scheduling frequency, as shown in Tables 1 and 2, the six sets of comparison cases (IEEE33+IEEE69) set in this scenario can run within 3 minutes (minimum of 8.555s), which can meet the real-time power grid scheduling frequency requirements. Thirdly, in terms of solving accuracy, as shown in Figure 11, the magnitude of the error in the simulation results of this scenario is strictly controlled within 10-6, which can ensure the requirement of power grid solving accuracy. (on the lines 568-580 of page 18) 3.2 Network reconfiguration ……
Figure 15. Voltage of different nodes. As shown in Figure 14, It is not difficult to find that the error is orders of magnitude, meeting the reconstruction requirements. It is worth noting that in this example, no active management equipment is added. For IEEE 33 nodes, all nodes have load values, so only equation (33) can be added to the radiation constraint. For some nodes in the IEEE69 system without load, equation (34) must be added to ensure that all nodes are connected and operate without islands and radiation. Based on the simulation results of this section, the three important parameters of computation time, error, and node voltage distribution are evaluated. Firstly, in terms of power grid stability, as shown in Figure 15, the voltage distribution of each node in the IEEE33 node network can meet the set range requirement of [0.94, 1.06], and the voltage distribution in multi-period is relatively uniform. Secondly, regarding power grid scheduling frequency, as shown in Table 3, the two sets of comparative cases (single-period and multi-period) ensure that the operating time is within 3 minutes (the shortest being 1.355s), meeting the requirements of real-time power grid scheduling frequency. Thirdly, in terms of solving accuracy, as shown in Figure 14, the error level of the simulation results in this scenario is strictly controlled within 10-9, which can well guarantee the accuracy requirements of power grid solving. (on the lines 605-615 of page 19) 3.3 ZIP load application …… Based on the simulation results of this scenario, calculation time, solution error, and node voltage distribution, are evaluated. Firstly, in terms of power grid stability, as shown in Figure 17, the voltage distribution of each node in the IEEE33 node network can meet the requirements of the [0.94, 1.06] range. Secondly, in terms of power grid dispatch frequency, the running time of this case is 5.394 seconds, which can meet the requirements of real-time power grid dispatch frequency. Thirdly, in terms of solution accuracy, as shown in Figure 19, the error level of the simulation result in this scenario is strictly controlled within 10-6, which can guarantee the requirement of power grid solution accuracy. Finally, regarding the accurate modeling of distribution networks, as shown in Figure 18, the introduction of ZIP loads can change the load demand, demonstrating the importance of active and detailed modeling of distribution networks. (on the lines 639-650 of page 21)
|

Reviewer 2 Report
The abstract needs an indication of the obtained results and their significance.
There are reference lumps in the text such as: [1-3] . Please eliminate this lump. After that please check the manuscript thoroughly and eliminate ALL the lumps in the manuscript. This should be done by characterising each reference individually. This can be done by mentioning 1 or 2 phrases per reference to show how it is different from the others and why it deserves mentioning. This is not just a formalism. Having reference lumps in the text casts a serious doubt of whether the authors have really read and understood the cited sources. If you do not characterise the references individually it matters little how they are formatted. What really matters is to have meaningful references and the requirement for individual characterisation aims exactly at that. Even the small lumps leave something unsaid and reduce the quality and impact of the paper. Note - just adding the author names is not a sufficient distinguishing characterisation of the references.
Besides that issue, the introduction section is well composed, easy to read and makes clearly what are the main problems in OPF area and what is the novelty of the current work. A small recommendation, if the authors would agree - the current introduction text emphasises too much the solution side of the modelling. It would be nice to make the issues from the problem area to stand out more. I leave it at the discretion of the authors to decide how appropriate this suggestion is.
Before proceeding to describe your chosen model and actions, please describe your scientific hypothesis, concepts and the relevant reasoning for choosing the particular modelling approach. This should be accompanied by an overall description of the procedure followed. A block diagram of the procedure would be also very useful.
The model description needs also more clarity. At the moment the provided equations are poorly explained - mainly listed and their variables named. For each equation please add a concise explanation including the reasoning for its definition, how does it work internally and how does it work within the model context.
In the conclusions, in addition to summarising the actions taken and results, please do explain their significance. It is recommended to use quantitative reasoning comparing with appropriate benchmarks, especially those stemming from previous work.
Author Response
Responses to Reviewer 2
(Notes: In the revised manuscript, the main changes are highlighted in yellow for clarity and easy reference.)
Comment 1: The abstract needs an indication of the obtained results and their significance.
|
Response 1: Thanks for the reviewer’s comment on our paper. We noticed that the indication of the obtained results and significances are lacked in the abstract. Based on your question, we added an indication in the abstract. Revised manuscript: Finally, three representative scenarios are used to evaluate the model accuracy and effectiveness. The results show that the proposed LSOCR-DOPF model can ensure calculation time within 3 minutes, voltage stability, and error control within 10-6 for all three applications. This method has strong practical value in the fields of active distribution network day-ahead dispatch, accurate modeling of ZIP load, and real-time operation. (on the lines 11-14 of Abstract, page 1)
|
Comment 2: There are reference lumps in the text such as: [1-3] . Please eliminate this lump. After that please check the manuscript thoroughly and eliminate ALL the lumps in the manuscript. This should be done by characterising each reference individually. This can be done by mentioning 1 or 2 phrases per reference to show how it is different from the others and why it deserves mentioning. This is not just a formalism. Having reference lumps in the text casts a serious doubt of whether the authors have really read and understood the cited sources. If you do not characterise the references individually it matters little how they are formatted. What really matters is to have meaningful references and the requirement for individual characterisation aims exactly at that. Even the small lumps leave something unsaid and reduce the quality and impact of the paper. Note - just adding the author names is not a sufficient distinguishing characterisation of the references.
Besides that issue, the introduction section is well composed, easy to read and makes clearly what are the main problems in OPF area and what is the novelty of the current work. A small recommendation, if the authors would agree - the current introduction text emphasises too much the solution side of the modelling. It would be nice to make the issues from the problem area to stand out more. I leave it at the discretion of the authors to decide how appropriate this suggestion is.
|
Response 2: Thanks for your valuable comment on our paper. As suggested, we have checked the whole paper, and reference lumps like [1-3] are fully avoided to achieve characterizing each reference individually. Besides, based on your helpful suggestions, we have supplemented the existing problems in the research field in the introduction section. It should be noted that some of the consecutive citations in the paper are due to the fact that the relevant authors divided the same research content into two parts to write separate papers. For example, the references "[13,14], " [16,17]", and "[13,14,16,17]" cited in this paper were written in two separate parts by the same author. All the corrections are marked in the revised manuscript. Revised manuscript: â‘ Recently, researchers have shown an increased interest in the active distribution network. The integration of various distributed generation, energy storage units, and active management devices has brought new challenges to the planning and operation of distribution networks [1], especially in the field of active management (AM) of distribution networks [2]. Ref. [3] analyzed three kinds of optimization problems of the smart grid: optimal power flow (OPF), unit commitment, and operation planning. Their essence is distribution network optimization, while having different optimization scale. (on the lines 34-41 of page 1) â‘¡ With the continuous development and maturity, commercial optimization software (GUROBI, CPLEX, MOSEK, etc.) has been widely used in distribution network reconstruction [29], reactive power optimization [30], distribution network planning [11], etc. (on the lines 122~125 of page 3) â‘¢ Additionally, although the OPF model of the active distribution network based on MISOCP has achieved a relatively high solution efficiency [21], strictly speaking, it is still a nonlinear model, and the efficiency of solving nonlinear second-order cone constraints will de-crease with the increase of the number of distribution network nodes. In order to achieve efficient solving of the OPF problem in future large-scale distribution networks, it is necessary to simplify the second-order cone programming constraints while en-suring the accuracy and efficiency of distribution network solving. Ref. [22] has proven that the second-order cone in dimension N, can be outer-approximated to an arbitrary accuracy by a polyhedral cone in an extended space. Polyhedral approximations are very powerful for solving MISOCP as the benefits of warm-starting for LPs can be utilized throughout the branch-and-bound algorithm [23]. This is particularly important for problems that can be formulated as MISOCP, such as the OPF problem in this paper. Meanwhile, Refs. [24,25] extended the constant power load model to a ZIP load model under a rectangular coordinate system and applied it to the solution of the OPF. (make the issues from the problem area to stand out more, on the lines 100~116 of page 3) ……
|
Comment 3: Before proceeding to describe your chosen model and actions, please describe your scientific hypothesis, concepts and the relevant reasoning for choosing the particular modelling approach. This should be accompanied by an overall description of the procedure followed. A block diagram of the procedure would be also very useful.
|
Response 3: We are very grateful for the kind advice of the dear reviewer. As suggested, we highlighted the important equations in the modeling process with a flowchart in Figure 2 and explained them in detail in the context, so that readers can intuitively understand the key modeling equations. In addition, we described in Figure 3 the linear constraints of the active management units incorporated into the LSOCR-DOPF model, to provide a comprehensive understanding of the model. All the corrections are marked in the revised manuscript. Revised manuscript: Meanwhile, most of the existing distribution network models only include power generation units and energy storage devices, and rarely consider reactive power compensation devices and other active management devices at the same time [26]. In addition, in the active distribution network planning and operation optimization model, discrete variables will inevitably appear with the active management devices considered, which turns the original problem into a mixed integer linear programming (MILP) [27]. With the continuous development and maturity, commercial optimization software (GUROBI, CPLEX, MOSEK, etc.) has been widely used in distribution network reconstruction [28], reactive power optimization [29], distribution network planning [11], etc. It is noteworthy that, the above studies are mostly limited to the traditional single-period static OPF category, while the actual optimization requires the overall coordination of multi-period, which is actually a dynamic optimal power flow (DOPF). Additionally, compared to the rectangular coordinate system, the current form in polar coordinates is more common. Therefore, in order to facilitate understanding, additional explanation is required for the entire model transformation process (from AC-OPF to LSOCR-DOPF): Equations (8) and (16) respectively embody the phase angle relaxation and second-order cone relaxation of LSOCR-DOPF, and Figure 2 depicts the schematic diagram of the two-step relaxation process. The non-convex feasible region of the original AC-OPF problem will be relaxed into a convex second-order cone feasible region after phase angle relaxation and second-order cone relaxation. Then, the convex feasible region of the second-order cone is further linearized into the convex feasible region of the integer programming by the polyhedral approximations. At this time, the optimal power flow problem in the original formulation has already been transformed into a convex optimization problem. Numerous studies, as demonstrated in Refs. [13,14,16,17], have substantiated the strict accuracy of the second-order cone relaxation (SOCR) approach for most distribution network structures, when the objective function is both a convex and strictly increasing function.
Figure 2. The Proposed modeling process of LSOCR for solving AC-OPF. (on the line 268 of page 7) ……
Figure 3. Proposed LSOCR-DOPF optimization flowchart. (on the line 431 of page 12) ……
|
Comment 4: The model description needs also more clarity. At the moment the provided equations are poorly explained - mainly listed and their variables named. For each equation please add a concise explanation including the reasoning for its definition, how does it work internally and how does it work within the model context.
|
Response 4: Thanks for your valuable advice on our paper. As suggested, we have revised the modeling process of LSOCR-DOPF in the main text, including the addition of references to classic equations cited in previous works and detailed explanations of our own designed equations and physical parameters used in the model. A brief explanation of the equations used in the modeling section of the active management units (CB, SVC, OLTC, etc.) is provided at the end of each respective equation.
Revised manuscript: …… Besides, and represents complex power, and . And branch complex impedance ; Set represents the set of all nodes in the network; In the traditional OPF, the voltage remains constant. If OLTC is installed, the voltage will change with the OLTC transformation ratio; E represents the collection of all branches in the network. There are substations, nodes and line branches in the network.
Figure 1. Structure of radial distribution network. 2.1.2 Basic OPF model based on BFM Generally, the basic model of optimal power flow based on branch power flow (BFM-OPF, Figure 1) is expressed as follows [14]: (1) s.t. (2) (3) (4) (5) (6) (7) where, and represent the active and reactive power injections, respectively, at each node; the branch represents the positive direction of flow direction from node to node ; is the collection of branch end nodes with as the head node, and is the collection of branch end nodes with as the end node; and denote the active and reactive power at the head node of the branch , respectively; and denote the active and reactive current flow in each branch; and correspond to the resistance and reactance of each branch, individually; and are the ground conductance and ground susceptance of node , separately; and are the upper and lower limits of node voltage, respectively; and are the upper and lower limits of branch current, respectively. From equations (1) to (7), it can be inferred that: 1) The optimization variables for OPF consist of node injection power , branch power flow , node voltage , and branch current , with the substation node voltage not considered as an optimization variable; 2) Equation (1) represents the objective function, which can be the minimization of network losses and substation node power purchases; 3) The well-known branch flow equation [30] is expressed by equations (2)-(3), while equation (4) represents the power calculation equation. Equations (5) and (6) respectively denote the safety constraint equations for branch currents and nodal voltages. Equation (7) introduces node-dependent constraints that are subject to change based on the employed model. …… (8) Substitute equation (8) into equations (2) ~ (6) to change the power flow constraint to the second-order cone constraint [31] as follows: …… With this, the SOCR-OPF (a typical Mixed-Integer Second Order Cone Programming, MISOCP) model is fully modeled. Among them, the second-order cone programming represented by equation (12) can be expanded and simplified as : (15) The equation (15) can be uniformly described as: (16) Ben-Tal and Nemirovski [22] show that equation (16) can be approximated by a system of linear homogeneous equalities and inequalities in terms of , and variables for . is a parameter of the polyhedral-relaxed approximation. The approximate expression of the polyhedron of the three-dimensional SOCR constraint (equation (16)) is: (17) where, represents the number of facets in a polyhedron. and are auxiliary variables. The polyhedral approximation given by (17) can be reduced by using the linear equality constraints to solve for in terms of and and then substitute out of the equation (17). The resulting system will only have linear inequality constraints in terms of , and the variables for . The error ( ) [22] of the polyhedral approximations of the three-dimensional second-order cone constraint (Equation (16)) is: (18) According to Equation (18), when K=11, the error is about . Therefore, the MISOCP model for active distribution network reconfiguration is approximately equivalent to the MILP model. At this point, the improved version of SOCR-OPF, namely the LSOCR-OPF model, is fully modeled. So far, the power flow constraint has been transformed from a nonlinear programming model to a MISOCP model composed of equations (9) ~ (14). Then, through the polyhedron approximation method which is used to linearize equation (16), the MISOCP model can be approximately converted into a MILP model for solution. ……
|
Comment 5: In the conclusions, in addition to summarising the actions taken and results, please do explain their significance. It is recommended to use quantitative reasoning comparing with appropriate benchmarks, especially those stemming from previous work.
|
Response 5: We appreciate the reviewer for pointing this issue on our paper. Due to the authors' negligence, in the original manuscript, we did not summarize the significances of the results in the conclusion section. Considering the purpose of engineering applications, we evaluated three key indicators (calculation time, solution error, and node voltage distribution) that are more relevant to the power distribution network. Additionally, we included a separate paragraph in the conclusion discussing future research challenges, which could further enhance the understanding of our findings.
Revised manuscript: The results demonstrate that LSOCR-DOPF meets the requirements of timely calcula-tion and accuracy in the fields of day-ahead optimal dispatch, real-time operation, and efficient control for active distribution networks. Additionally, the results highlight the importance of accurately modeling ZIP load in active distribution networks. The results show that the strategy proposed in this paper can meet the engineering standards for typical application scenarios of distribution networks: on one hand, the calculation time is controlled within 3 minutes, which satisfies the requirement of the general power grid scheduling refresh rate and ensures fast solution of OPF problems in future large-scale active distribution networks; on the other hand, the calculation result error is controlled within 10-6, effectively avoiding catastrophic consequences caused by calculation errors in previous work; moreover, the voltage distribution of each node in the distribution network meets the condition constraints [0.94, 1.06] set by authors, which can achieve the optimization goal of reducing network loss. It is noteworthy that the LSOCR-DOPF model satisfies the requirements of calculation time and accuracy in the fields of daily scheduling, real-time operation, and control of active distribution networks, demonstrating strong practical application value, and the case study suggests that considering the ZIP model for fine simulation of active distribution networks is also of great significance. (on the lines 657-670 of page 21)
|
